# Individualizing glioma radiotherapy planning by optimization of a data and physics-informed discrete loss

Michal Balcerak [1] ✉, Jonas Weidner[2,3], Petr Karnakov[4], Ivan Ezhov[2], Sergey Litvinov[4,5], Petros Koumoutsakos [4], Tamaz Amiranashvili [1,2,6], Ray Zirui Zhang[7], John S. Lowengrub[7,8], Igor Yakushev [9], Benedikt Wiestler [3,10,11] & Bjoern Menze [1,11]

Brain tumor growth is unique to each glioma patient and extends beyond what is visible in imaging scans, infiltrating surrounding brain tissue. Understanding these hidden patient-specific progressions is essential for effective therapies. Current treatment plans for brain tumors, such as radiotherapy, typically involve delineating a uniform margin around the visible tumor on pre-treatment scans to target this invisible tumor growth. This "one size fits all" approach is derived from population studies and often fails to account for the nuances of individual patient conditions. We present the Glioma Optimizing the Discrete Loss (GliODIL) framework, which infers the full spatial distribution of tumor cell concentration from available multi-modal imaging, leveraging a Fisher-Kolmogorov type physics model to describe tumor growth. This is achieved through the newly introduced method of Optimizing the Discrete Loss (ODIL), where both data and physics-based constraints are softly assimilated into the solution. Our test dataset comprises 152 glioblastoma patients with pre-treatment imaging and post-treatment follow-ups for tumor recurrence monitoring. By blending data-driven techniques with physics-based constraints, GliODIL enhances recurrence prediction in radiotherapy planning, challenging traditional uniform margins and strict adherence to the Fisher-Kolmogorov partial differential equation model, which is adapted for complex cases.

Gliomas are the most common primary brain tumors in adults.[1,2] Commonly used treatment strategies include surgery, chemotherapy, and radiotherapy. Despite advances in understanding the biological basis of these tumors and the multi-modal combination of therapies, the prognosis of glioma patients, in particular those with glioblastoma

(WHO-CNS grade 4)[3], remains dismal. A key challenge for more successful therapy of glioma patients is the infiltrative tumor growth pattern. Already at initial diagnosis, glioma cells have invaded the surrounding brain parenchyma well beyond the tumor margins visible on conventional imaging. To account for this otherwise invisible tumor

[1]Department of Quantitative Biomedicine, University of Zurich, Zurich, Switzerland. [2]Department of Computer Science, Technical University of Munich, Munich, Germany. [3]Munich Center for Machine Learning (MCML), Munich, Germany. [4]Computational Science and Engineering Laboratory, Harvard University, Cambridge, MA, USA. [5]Computational Science and Engineering Laboratory, ETH Zurich, Zurich, Switzerland. [6]Visual and Data-Centric Computing, Zuse Institute, Berlin, Germany. [7]Department of Mathematics, University of California, Irvine, CA, USA. [8]Department of Biomedical Engineering, University of California, Irvine, CA, USA. [9]Department of Nuclear Medicine, Technical University of Munich, Munich, Germany. [10]Chair of AI for Image-Guided Diagnosis and Therapy, TUM School of Medicine and Health, Munich, Germany. [11]These authors contributed equally: Benedikt Wiestler and Bjoern Menze. ✉e-mail: michal.balcerak@uzh.ch

growth, both North American and European guidelines for radiotherapy planning define standard, uniform safety margins around the resection cavity and/or remaining tumor based on conventional Magnetic Resonance Imaging (MRI), which highlights areas affected by edema alongside regions of actively growing and necrotic tumor tissue[4,5]. Despite many efforts, truly tailoring radiotherapy to an individual patient's tumor's spread is an unmet clinical need in neurooncology[6].

In current clinical practice, radiotherapy planning for glioblastoma patients typically employs a uniform safety margin of 1.5 cm[7] around the visible tumor core identified in preoperative MRI scans. This standardized approach aims to encompass both the detectable tumor and potential microscopic extensions that are not visible on imaging. However, this uniform margin does not account for the highly heterogeneous nature of tumor infiltration, variations in individual brain anatomy, or the presence of anatomical barriers that can influence tumor spread. Consequently, the uniform margin may either over-treat healthy brain tissue, leading to unnecessary side effects, or under-treat regions where the tumor has infiltrated beyond the prescribed margin, increasing the risk of early recurrence. These shortcomings highlight the urgent clinical motivation to seek more personalized and precise radiotherapy planning methods that can adapt to the unique tumor dynamics and anatomical conditions of each patient.

Computational modeling has the potential to improve the definition of radiotherapy volume[8,9], particularly through the use of partial differential equations (PDE) to simulate the nuanced progression of tumors. This approach, exemplified by models like the Fisher-Kolmogorov (FK) equation, enables the prediction of spatial tumor cell distribution by accounting for migration and proliferation. Although more intricate diffusion-reaction models introduce the complexity of various tumor cell states, such as necrotic and proliferative, they also require extensive data for precise parameter estimation. Incorporating multimodal imaging, including Fluoroethyl-L-Tyrosine Positron Emission Tomography (FET-PET), our framework aims to refine predictions of tumor infiltration patterns. FET-PET provides metabolic information[10,11] that, when combined with structural data from MRI, offers a complete view of tumor behavior. This integration not only parallels dose-boosting strategies in targeting areas of high metabolic activity but also underscores the potential of using complex multimodal imaging data to inform models that predict tumor growth and spread with greater precision.

Existing approaches to personalizing tumor growth models require solving the inverse problem, i.e., inferring the growth model parameters that provide an optimal fit to the clinically observed tumor on images[12–20]. Given that we only have a single time point of data when initiating treatment, the growth models employed must be exceedingly simple, leading to significant mismatch between model parameters and image observations, and requiring strong assumption about initial conditions. Without this simplicity, the problem becomes highly ill-posed, and accurate retrieval of parameters becomes unattainable. In addition, traditional methods, e.g., those based on Monte Carlo sampling[21], have a severe limitation, namely, long computational time to perform parametric inference. Both the computational time and simplistic tumor growth equations severely limit clinical applicability of such models for radiotherapy planning.

Recently, deep learning methods were introduced to address the computational time issue of tumor cells inference[22–28]. While these learnable methods offer improvements in computational efficiency, their current lack of robust error control and solution accuracy poses significant challenges in medical applications, where the utmost precision is required for treatments and patient care. Until these issues are adequately addressed, the use of such models in critical medical decisions remains limited.

Physics-Informed Neural Networks (PINNs) emerge as a middle ground, aiming to strike a balance between the rigidity of PDE models, initial conditions of the tumor cells spatial distribution, and the flexibility of data-driven approaches. They embed physical laws in the form of differential equations directly into the architecture of neural networks[29]. Theoretically, this allows for more reliable and physically meaningful predictions by using the neural network to approximate the solution to the PDE and adapt the initial condition accordingly to soft imposed assumptions. However, the practical application of PINNs in a clinical setting is currently challenging due to computational inefficiencies. Modifying a single weight in these often densely connected networks can have a widespread, non-local impact on their output, making calibration a notoriously difficult task, limiting its clinical applicability. Moreover, although the PDE residual serves as a penalty term in PINNs, its testing is confined to a restricted number of points rather than being evaluated globally across the entire computational domain. Hence, while PINNs offer a promising avenue for model personalization, they currently fall short in terms of computational feasibility for real-world applications.

In summary, three key challenges must be addressed to facilitate the successful clinical translation of computational tumor growth models: (i) enhancing computational efficiency to enable timely and practical use in a clinical setting, (ii) establishing error control to ensure safe clinical application, and (iii) introducing the flexibility to adapt models beyond basic mathematical frameworks, thereby capturing the complexities of tumor growth more accurately. This necessitates a balance between adhering to the growth model and accurately reflecting the tumor observed in clinical practice.

This work introduces GliODIL, an optimization framework designed for estimating tumor cell concentrations and migration pathways surrounding visible tumors in MRI and PET scans, as depicted in Fig. 1. Our approach uniquely integrates traditional numerical methods with data-driven paradigms, providing a more comprehensive insight into tumor behavior. In addition to its adaptive capabilities. GliODIL builds upon our previous work on the ODIL technique[30,31] which significantly enhances computational speed compared to traditional PINN architectures. By utilizing consistent and stable discrete approximations of the PDEs, employing a multi-resolution grid, and leveraging automatic differentiation, we achieve computation times suitable for clinical applications such as radiotherapy planning.

Diverging from conventional glioma models, which primarily adjust parameters within predetermined PDE frameworks, our GliODIL methodology uniquely refines both the parameters and the discretized solutions. This refinement process involves optimizing a cost function that seamlessly integrates the growth equations in their discrete form with empirical data, treating these elements as soft constraints. This approach not only enhances computational efficiency by streamlining the optimization process but also incorporates error control mechanisms. These mechanisms assess the model's fidelity by calculating the physics residual error and evaluating how closely our assumptions align with the tumor progression reflected in the clinical data. Furthermore, GliODIL introduces the necessary flexibility to move beyond basic mathematical models, thereby more accurately capturing the intricate realities of tumor growth and progression.

In this work, we show that GliODIL effectively addresses the outlined challenges through dedicated experiments on clinical data. Our test dataset comprises 152 glioblastoma patients, including 58 with pre-treatment FET-PET imaging alongside MRI, and post-treatment MRI follow-ups for comprehensive tumor recurrence monitoring and model validation. These follow-ups occur within 1 to 12 months after treatment, providing a broad spectrum to assess GliODIL's performance in real-world clinical scenarios. By making this dataset publicly available, we validate GliODIL's efficacy in enhancing tumor recurrence coverage and contribute to the broader pursuit of personalized treatment strategies in oncology.

**a** Input Patient Data and Preprocessing

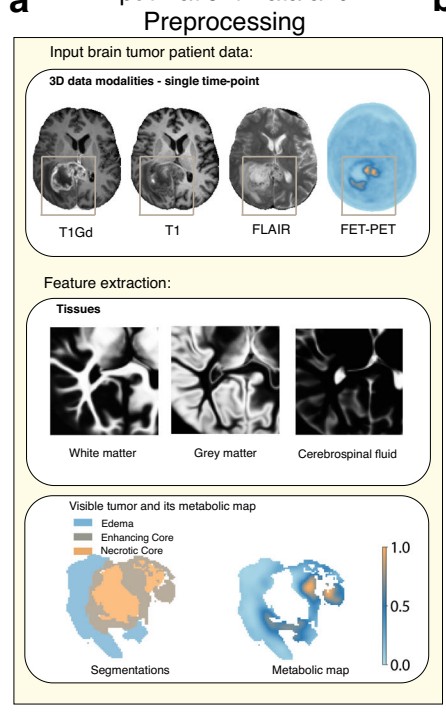

**b** Physics-Informed Constraints: Tumor Growth Model and Imaging Model

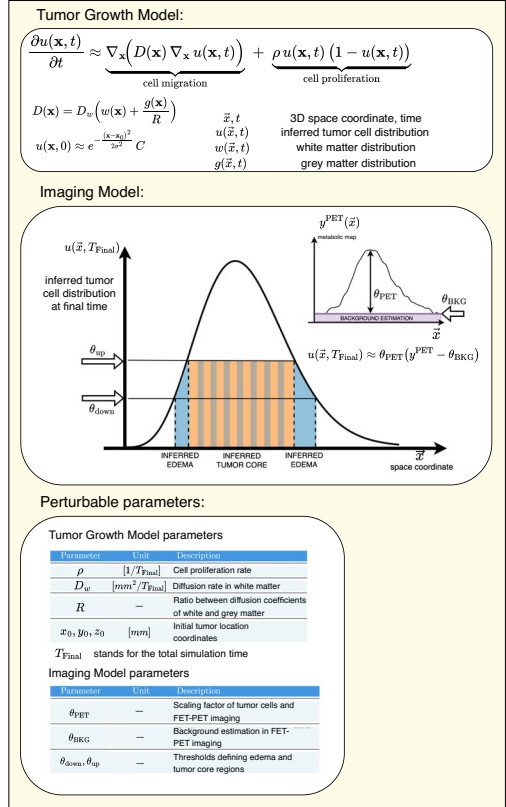

**c** GliODIL: Optimization of Data and Physics-Informed Discrete Loss

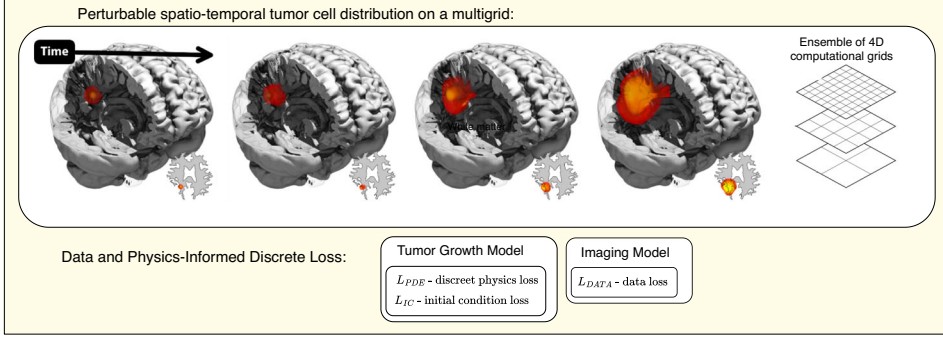

**d** Output Patient-Specific Tumor Distribution and Radiotherapy Plan

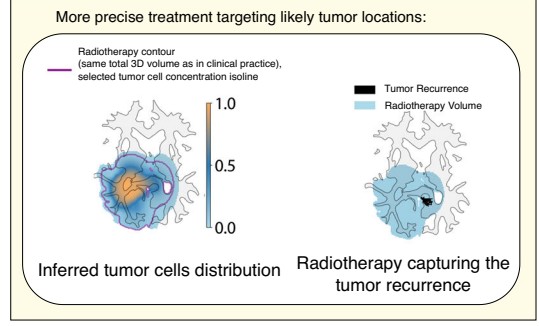

## Results

To provide context for our experimental findings, we begin with a concise overview of the GliODIL pipeline (Fig. 1), illustrating how input data, prior knowledge of tumor growth, and PDE-based constraints combine to yield patient-specific predictions of tumor evolution and inform radiotherapy planning.

As shown in Fig. 1a, GliODIL operates on anatomical and metabolic information derived from MR and FET-PET imaging. Tissue extraction using atlas registration[21] and automatic tumor segmentation[32] delineate the tumor's boundaries (edema, enhancing core, and necrotic core), providing the necessary spatial context for subsequent analysis.

**Fig. 1 | GliODIL overview. a** Multi-modal patient data comprising MR and PET imaging. Tissue extraction using atlas registration[21] and automatic brain tumor segmentation[32] are performed to define the tumor boundaries and microenvironments. Automatically segmented tumor regions include three components:(i) edema, characterized by tissue swelling due to fluid accumulation; (ii) enhancing core, indicative of active tumor growth and characterized by vascular leakage, and (iii) necrotic core, showing tissue death due to hypoxia or nutrient deprivation. Corresponding FET-PET scans provide metabolic insight, further aiding in accurate tumor delineation. **b** Prior knowledge about the tumor growth process and the imaging signatures of tumor cells. The physics of tumor growth is described by a partial differential equation (PDE), while the relationship between tumor cells and the available imaging data is modeled through the Imaging Model. **c** Spatio-

temporal progression of a tumor within patient anatomy. Calculation of PDE's residual $L_{PDE}$ and single focal initial condition $L_{IC}$. Unknown fields are stored on a 4-dimensional multi-resolution grid. Optimization utilizes automatic differentiation of each gridpoint and is guided by the loss function. Since we model the progression based on a single time-point input data, the growth parameters are being resolved up to a timescale. Calculation of discrepancy between patient's tumor characteristics $L_{DATA}$ and proposed by GliODIL tumor cell-distribution at the final time-point. **d** GliODIL outputs. The framework successfully infers the complete distribution of tumor cells, facilitating the development of a radiotherapy plan. This plan effectively covers areas of tumor recurrence identified in post-operative data, while maintaining the total radiotherapy volume in line with standard clinical guidelines.

---

Figure 1b highlights how physics- and imaging-based constraints are integrated. Specifically, the Tumor Growth Model (section Tumor Growth Model) uses a Fisher-Kolmogorov Reaction-Diffusion (FK) PDE to capture cellular diffusion and proliferation, and the Imaging Model (section Imaging Model) links these tumor cell distributions to the observed MR and PET data.

As depicted in Fig. 1c, the model estimates the tumor's 4D evolution—storing unknown fields in a multi-resolution grid—while minimizing a comprehensive loss function (section Final Loss Function). This includes enforcing PDE constraints ($L_{PDE}$), specifying focal initial conditions ($L_{IC}$), and capturing alignment with patient imaging ($L_{DATA}$). The ODIL framework (section Optimizing a Discrete Loss (ODIL)) guides this optimization, balancing data fidelity against physics-based regularization.

Finally, Fig. 1d illustrates the outcome of this optimization. GliODIL infers the full tumor cell distribution within the brain and provides a basis for improved radiotherapy planning. By preserving standard clinical dose thresholds while covering areas at high risk of recurrence, GliODIL yields treatment volumes comparable to conventional margins but with better targeted coverage of complex tumor shapes.

After introducing this streamlined workflow, we evaluate GliODIL's performance on synthetic data (section Full Spatial Tumor Cell Distribution Inference)—enabling parameter fine-tuning and robust testing—and subsequently on real tumor cases from 152 patients (section Clinical Dataset Validation: Radiotherapy Planning and Recurrence Coverage). These analyses compare GliODIL against standard practices (the "Standard Plan") and multiple baseline models (section Baselines), employing Dice score, FET-PET correlation, and Recurrence Coverage (section Evaluation Metrics) to demonstrate GliODIL's predictive accuracy. GliODIL is the only model that consistently outperforms the Standard Plan.

The study is organized as follows: In section "Full Spatial Tumor Cell Distribution Inference", synthetic data is used to evaluate model performance under varied tumor conditions, allowing for parameter fine-tuning against a known ground truth. These conditions involve cases of individual tumors and localized multi-focal tumors, demonstrating the model's robustness in handling noise and scenarios resembling post-resection conditions. Subsequently, in section "Clinical Dataset Validation: Radiotherapy Planning and Recurrence Coverage", GliODIL is applied to and compared against multiple baselines in real tumor cases from 152 patients (discussed in section Clinical Dataset) to assess its accuracy in inferring tumor growth parameters that depict patient-specific scenarios, benchmarking these findings against established methods described in section "Baselines". For radiotherapy planning, we compare our results with the standard clinical practice of applying uniform safety margins around preoperative MRI scans, referred to as the Standard Plan (Clinical Target Volume, CTV).

Our analysis integrates cell concentrations measured directly from GliODIL and outcomes from forward PDE simulations using parameters estimated by GliODIL, denoted as PDE$_{GliODIL}$. To enhance

GliODIL's efficiency, we implement a personalized initial guess for tumor cell distribution, detailed in section "Initial Guess".

## Full spatial tumor cell distribution inference

Our main modeling assumption is that tumor growth according to a given PDE is controlled by the loss $L_{PDE}$ and that it starts from a single focal seed point, controlled by the initial condition $L_{IC}$ loss. Both of these loses are formally introduced in section "Tumor Growth Model". We want to test the applicability of these assumptions to more complex, real-world scenarios like localized multi-focal tumors that break the modeling assumptions. We considered relaxing both assumptions to capture such complex scenarios. As the initial condition of the tumor is uncertain and may not necessarily begin from a single focal point, we assign a low importance to it in our GliODIL solution. However, we experiment with the $L_{PDE}$ importance governed by a weighing parameter $\lambda_{PDE}$ in the final loss.

To add an additional layer of realism, we introduce noise into our synthetic FET-PET data using a random Markov field and partial volume effects around necrotic regions to study the robustness of our solution to imperfect data acquisitions, a common challenge in clinical application.

We report the results from 100 synthetic patients, half with single focal tumors and the remaining half with multi-focal. The outcomes of this experiment are illustrated in Fig. 2. In Fig. 2e we observe that both high $\lambda_{PDE}$ (strong adherence to the equation) and low $\lambda_{PDE}$ (overfitting to the data) are sub-optimal for complex tumors and the $\lambda_{PDE} = 1$ used in GliODIL aligns the closest with the ground truth.

The decision to set $\lambda_{PDE} = 1$ represents a balanced choice, enabling accurate inference of single focal tumors (see Fig. 2a–c) while also capturing a significant portion of the multi-focal ground truth, as shown in Fig. 2a, e. This setting of $\lambda_{PDE} = 1$ will be consistently applied in our subsequent experiments involving real patient data. Notably, the balance between adherence to the data and physics priors was determined using noisy synthetic data with broken assumptions rather than patient data. This approach prevents partial dilution of the validation dataset. If additional data becomes available, some recurrence cases could potentially be used to refine the calibration of $\lambda$ values.

For real patient data, we visualize two representative cases and present average results with their standard deviations in Fig. 3. For a detailed explanation of performance metrics, see section "Evaluation Metrics". Examining models that strictly adhere to tumor growth models' PDE, one observes that both PDE$_{GliODIL}$ and PDE$_{CMA-ES}$ explain the data at a comparable level. In contrast, PDE$_{LMI}$, despite its much faster inference, performs noticeably worse.

Regarding data explanation, as measured by Dice scores and PET signal correlation, GliODIL surpasses all examined PDE solutions. Better fit to the data can be attributed to GliODIL's ability to balance between adhering to the PDE on which it is being regularized on and effectively explaining the data. This performance indicates that its inferred tumor cell distributions could more accurately mirror actual conditions. However, for real patients, unlike in synthetic experiments

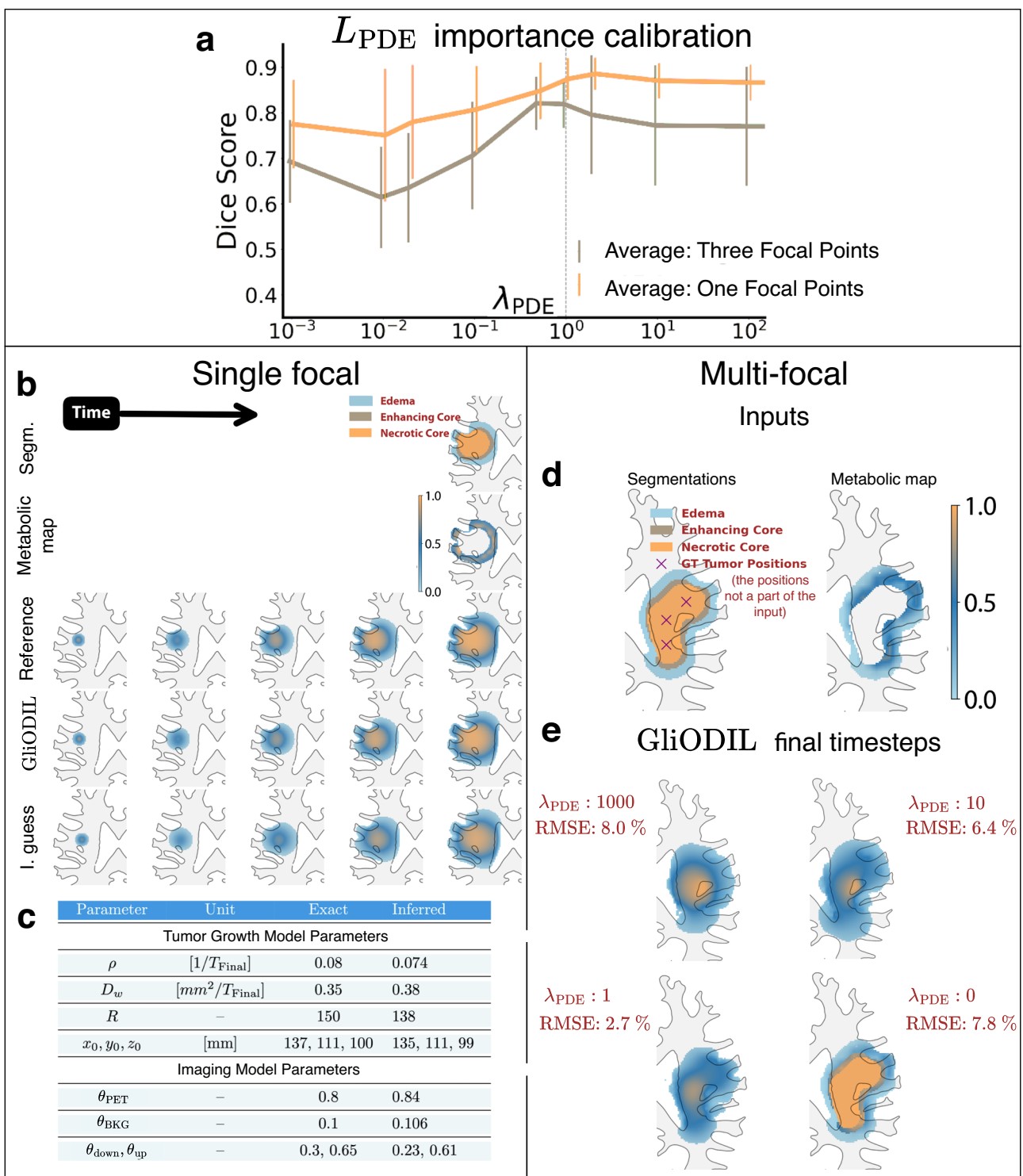

**Fig. 2 | Calibration of PDE loss weight in synthetic dataset experiments for tumor segmentation. a** Illustration of GliODIL's synthetic results for localized multi-focal and single focal tumors. In these simulations, all loss weights remain constant except for the $L_{PDE}$ weight, $\lambda_{PDE}$. A value of $\lambda_{PDE} = 1$ was selected to strike a balance in the model between fit to the data and the tumor growth equations. Dice scores were calculated at 10% tumor cell concentration, which is outside of the segmented region (edema at least at 20%). Data are presented as mean values ± standard deviation of $n = 50$ samples (synthetic patients) in each group. **b** The first and second rows represent a segmentation and a synthetic FET-PET that serve as input to GliODIL. Comparison: a forward run with ground truth parameters, tumor cells distribution inferred by GliODIL, and an initial guess (I. guess) that serves as a starting point of the optimization. **c** Comparison of exact and inferred parameters. **d** Input for the multi-focal experiment. **e** Tumor cell distribution of GliODIL with progressively relaxed $\lambda_{PDE}$. Root Mean Square Error (RMSE) is calculated for low-cell tumor concentration (at 10%), meaning tumor cell distribution outside of visible tumor on MRI.

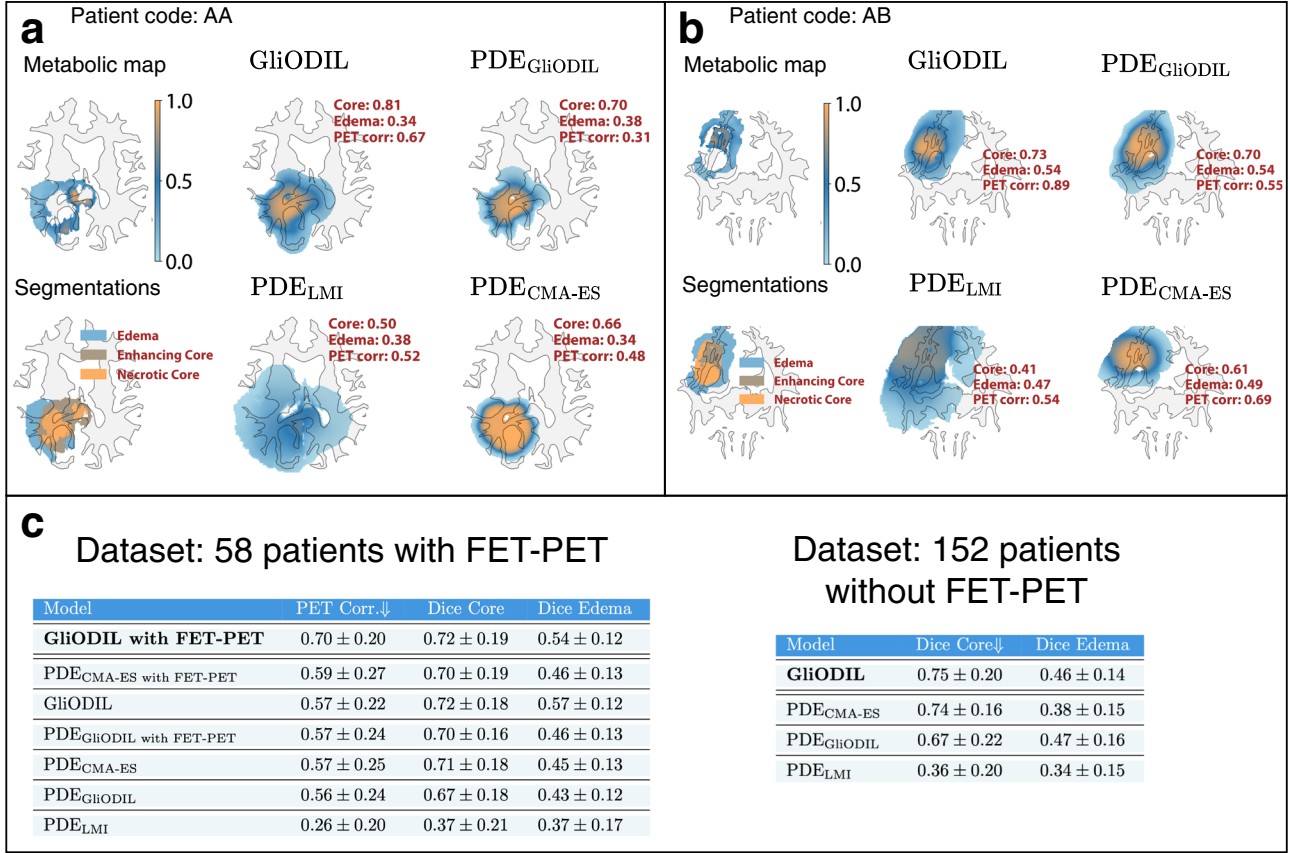

**Fig. 3 | Tumor cell concentration inference in real patient data: comparative analysis of predictive models. a**, **b** Comparison of tumor cell density predictions from various models with corresponding data inputs. Threshold segmentation values for Dice scores for each model are determined through a grid search, since the LMI model does not infer them. **c** Average data-fit scores for each model.

detailed in section "Full Spatial Tumor Cell Distribution Inference", we lack ground truth to directly substantiate these claims, and the results might be overfitted through the data-driven term (as in Fig. 2e for $\lambda_{PDE} < 1$). To validate GliODIL's performance, we conducted a downstream task with direct implications for clinical applications. In the following section "Clinical Dataset Validation: Radiotherapy Planning and Recurrence Coverage", we demonstrate that GliODIL leads to more effective radiotherapy plans regarding tumor recurrence coverage.

## Clinical dataset validation: radiotherapy planning and recurrence coverage

The primary metric for evaluating the model's efficacy is its accuracy in predicting tumor recurrence within the post-surgical radiation volume. The metric does not account for factors such as the extent of surgical resection or the impact of the radiotherapy that was administrated already to the patient. Nevertheless, it offers valuable insights into the model's potential to inform personalized radiotherapy planning by identifying tumor cell distribution beyond visible margins. This is particularly relevant for glioblastoma, where recurrences often occur adjacent to the resection cavity. We introduce a critical metric, Recurrence Coverage [%], detailed in section "Evaluation Metrics". This metric quantifies the percentage of follow-up MRI-detected recurrences, segmented and encompassed within a plan's radiation target. To ensure a fair comparison between the clinical practice of applying uniform safety margins (1.5 cm around the tumor core, adjusted for brain boundaries) and our GliODIL model's outputs, we ensured that the total radiotherapy volume, as represented in the 3D volume of treatment plans, remained constant across all models for each patient.

This consistency in radiation volume is crucial when interpreting the comparative figures. In Fig. 4, we illustrate both the clinical margins plan (using distance isolines) referred here as the Standard Plan and our GliODIL plans (using tumor cell concentration isolines). Our later discussed findings indicate that GliODIL outperforms all studied PDE models, highlighting the advantages of loosening the stringent PDE constraints found in conventional forward PDE simulations. This flexibility is particularly beneficial in radiotherapy planning, aiming to accurately pinpoint likely tumor locations by striking a delicate balance between empirical data and tumor growth equations. In demonstrating the contrast between models adhering strictly to tumor growth PDEs and GliODIL, Fig. 4 reveals that although PDE_GliODIL might surpass other PDE-strict methods in Recurrence Coverage, it faces challenges in complex tumor scenarios where PDEs inadequately capture the reality, occasionally missing certain tumor recurrences. Conversely, GliODIL effectively adjusts for equation discrepancies by integrating additional tumor cells in areas with high PET signal intensity via its data-driven component, thereby significantly improving recurrence coverage.

Our main analysis encompasses two distinct definitions of tumor recurrence: a broad definition including edema, enhancing core, and necrotic core, and a more specific definition aligned with current RANO guidelines[33] focusing only on the enhancing core. Figures 5 and 6 present the mean and standard deviation of Recurrence Coverage for these definitions. Furthermore, we conduct a patient-specific comparison with the Standard Plan, classifying a model as "Better" if it provides higher coverage for an individual patient and "Worse" if it falls short of the Standard Plan's coverage. In instances of equal coverage (e.g., both achieving 100%), we label the outcome as

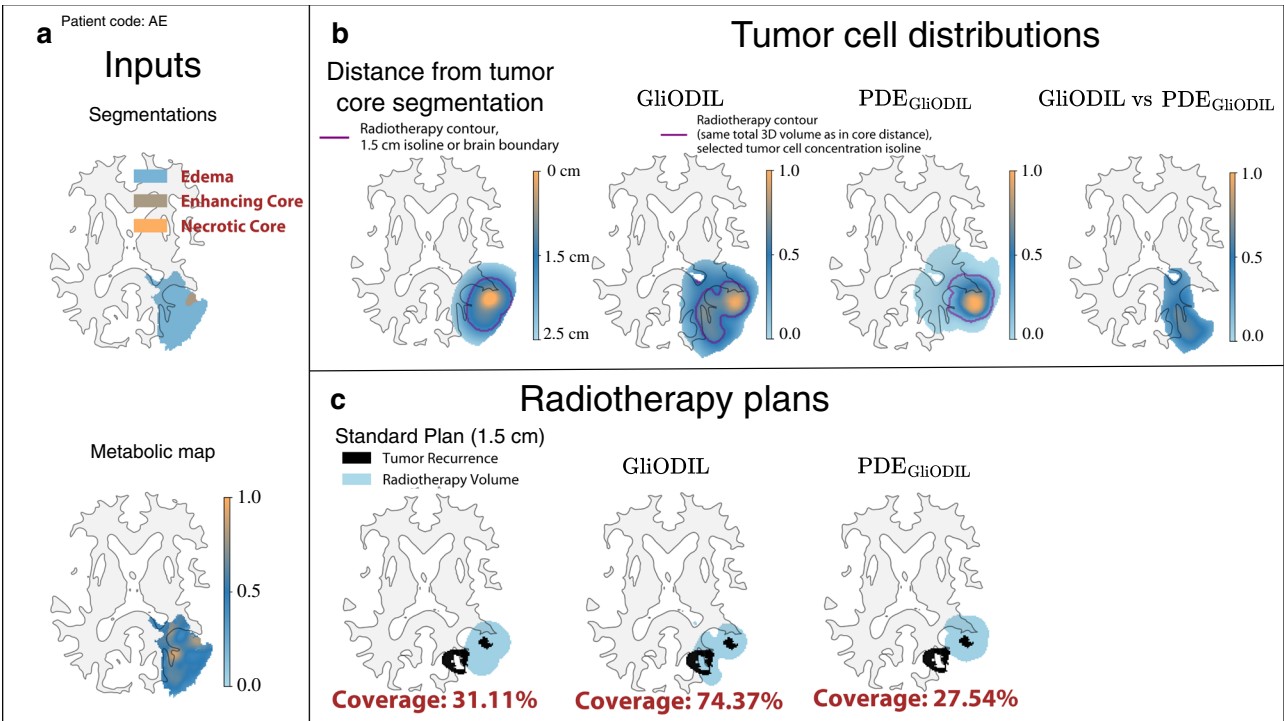

**Fig. 4 | Illustration of radiotherapy planning: uniform distance margin (Standard Plan) vs GliODIL vs. PDE solution. a** Model inputs including the FET-PET metabolic map and tumor segmentation data. **b** Illustration of the distance from the tumor core segmentation and its 1.5 cm isoline, adjusted for brain boundaries. This serves to define the Standard Plan and compare tumor cell distributions for both GliODIL and PDE$_{GliODIL}$, ensuring equal total 3D volumes across plans. Also shown is the absolute difference in distribution between GliODIL and PDE$_{GliODIL}$, exceeding 20%. **c** Visualization of radiotherapy plans including the Standard Plan, GliODIL, and PDE$_{GliODIL}$.

Equal. The comparison outcomes, including average results, are depicted in Figs. 5c and 6c.

Noticeable disparities are evident between the outcomes from any segmentation recurrence and those from enhancing core recurrence only. First, let's examine the results from radiotherapy targeting any segmentation recurrence. GliODIL demonstrates superior performance over uniform margin plans in most instances. Specifically, for the 58 patients with available FET-PET scans, the average Recurrence Coverage improved from 70.04% to 72.94%. For the 152 patients without FET-PET scans or where the scans were not incorporated into the model, the coverage increased from 63.59% to 67.80%. These improvements translate to a 35% and 38% difference in patient outcomes favoring GliODIL over the standard approach for the respective datasets.

Figure 5c reveals improvements, with the percentage of patients benefiting from the treatment increasing from 21% to 35% when comparing GliODIL, enhanced with FET-PET imaging, to the Standard Radiotherapy Plan with a 1.5 cm margin. This emphasizes the value of integrating FET-PET modality into flexible models capable of utilizing multi-modal data. Conversely, incorporating this modality into models rigidly conforming to a basic predefined PDE family yields statistically insignificant changes in performance.

To demonstrate scenarios where GliODIL excels, we present visualizations for two patients in Fig. 5a, b, showcasing recurrences located beyond the pre-operative visible tumor boundaries, thus eluding capture by the Standard Plan due to their position outside the 1.5 cm margin. GliODIL's integration of PET imaging and its regularization based on PDEs, which encode our knowledge about tumor migration paths, enables the model to capture some of these distant recurrences. This leads to consistently higher Recurrence Coverage. Traditional approaches for predicting tumor cell distributions involve estimating PDE parameters through inversion and conducting forward simulations within the patient's anatomy, specifically PDE$_{CMA-ES}$ and PDE$_{LMI}$ techniques. These methods have shown performance on par with the uniform margins defined in the Standard Plan. Additionally, the GliODIL method is characterized by reduced result variance.

In the analysis of enhancing core recurrence in follow-up scans, the hierarchy observed in the any segmentation recurrence study largely remains, with the exception that the Standard Plan demonstrates improved performance, outshining all models strictly based on PDE approaches. Instances of equal outcomes, as depicted in Fig. 6c, d, become more prevalent, indicating that both compared methods often achieve 100% coverage.

GliODIL stands out as the only model consistently surpassing the Standard Plan. This leads to a marked increase in coverage improvement, from benefiting 5% of patients to 19%. The conclusion drawn is that it's considerably more challenging for models to surpass the uniform margin around the pre-operative visible tumor. This difficulty arises because recurrences to the enhancing core typically occur closer to the resection cavity, rendering the migration paths, which these models leverage, less critical in this context compared to any segmentation recurrence. Any segmentation recurrences frequently occur in regions farther away, where anatomical features and topological barriers—factors not accounted for by a uniform margin—play a critical role. GliODIL, especially when incorporating FET-PET (see Fig. 5), leverages additional information to refine predictions while maintaining a strong alignment with pre-operative data. As illustrated in the figure, this approach demonstrates improved recurrence coverage compared to the standard margin-based method.

It's important to note that the high standard deviations in the Recurrence Coverage column reflect the inherent complexity of predicting tumor recurrences, which can vary significantly in difficulty from case to case. Despite this natural variance, the averages in the Recurrence Coverage column are a reliable predictor of the

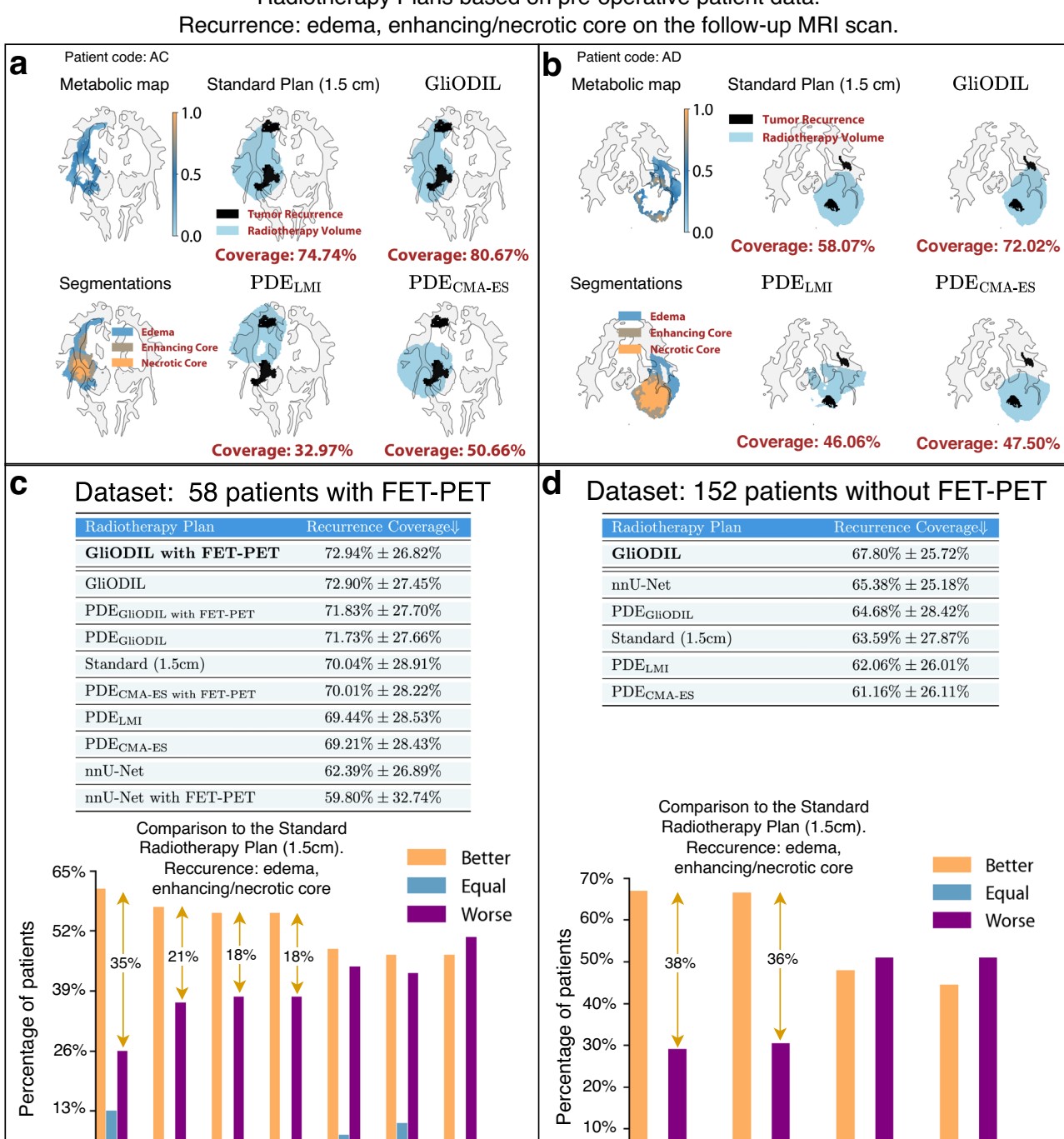

**Fig. 5 | Recurrence coverage analysis of edema, enhancing core, and necrotic core in real patient radiotherapy plans. a, b** Recurrence coverage of selected volume radiotherapy plans. All radiotherapy plans have the same total volume. Output tumor cell distribution thresholds found through a grid search to match the Standard Plan volumes. **c, d** Average Recurrence Coverage and direct patient-by-patient comparisons to the Standard Plan. Data are presented as mean values ± SD. Source data are provided with this paper.

effectiveness of each planning method, as the hierarchy of the recurrence scores translated to the direct comparisons with the Standard Plan.

To support future advancements, we present a case-by-case comparison with the Standard Plan, highlighting instances where

GliODIL performs suboptimally in Fig. 7. In Fig. 7b, incorporating brain fiber-related diffusion (as in ref. 34) into the PDE constraint could potentially improve performance, as tumor recurrence appears to follow the orientation of brain fibers. In Fig. 7c, the tumor's multifocal characteristics exceed the local multifocal scope within which GliODIL

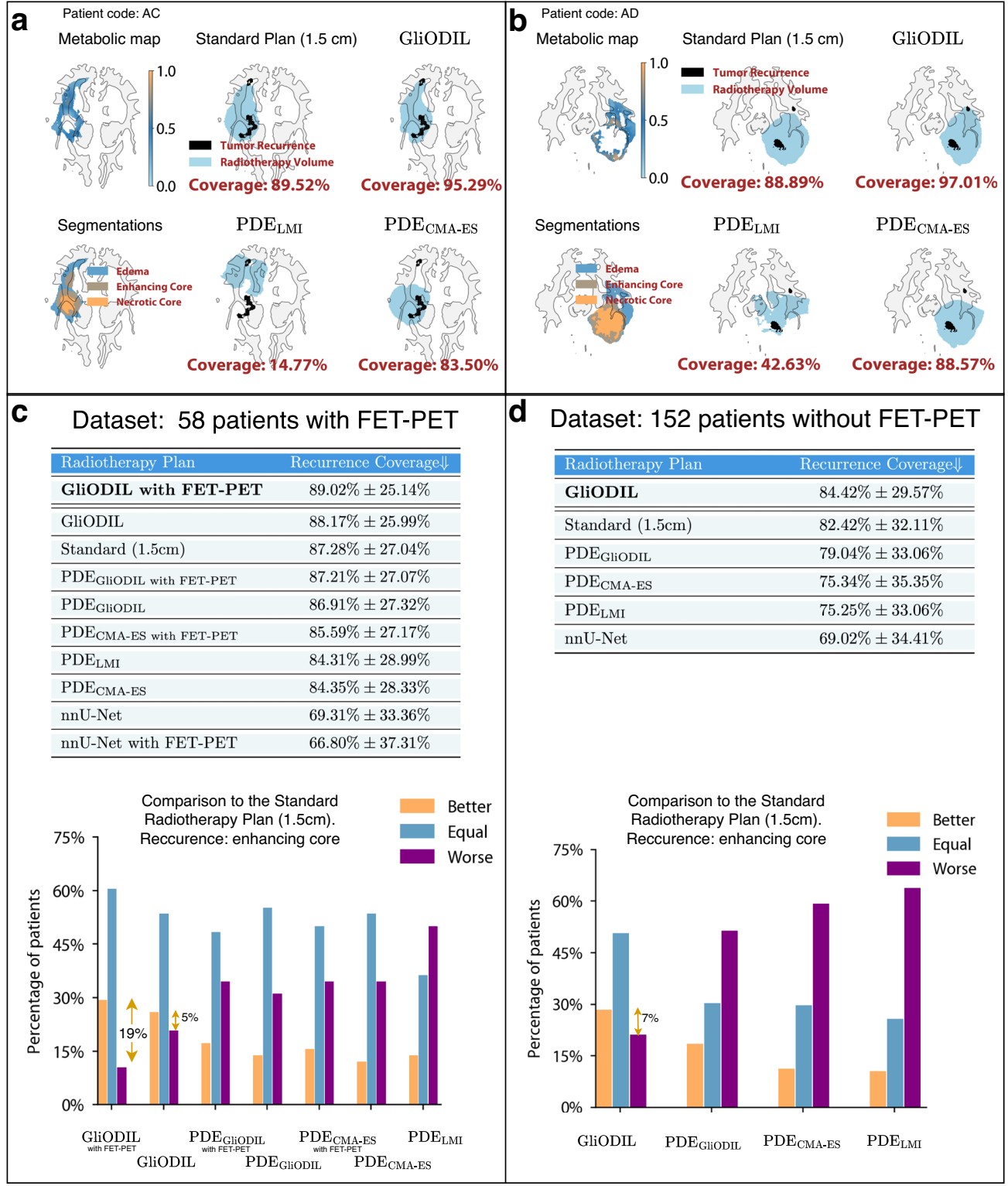

**Fig. 6 | Recurrence coverage analysis of the enhancing core in real patient radiotherapy plans. a, b** Recurrence coverage of selected volume radiotherapy plans. All radiotherapy plans have the same total volume. Output tumor cell distribution thresholds found through a grid search to match the Standard Plan volumes. **c, d** Average Recurrence Coverage and direct patient-by-patient comparisons to the Standard Plan. Data are presented as mean values ± SD. Source data are provided with this paper.

Case-by-case comparison with the Standard Plan.
Recurrence: edema, enhancing/necrotic core on the follow-up MRI scan.

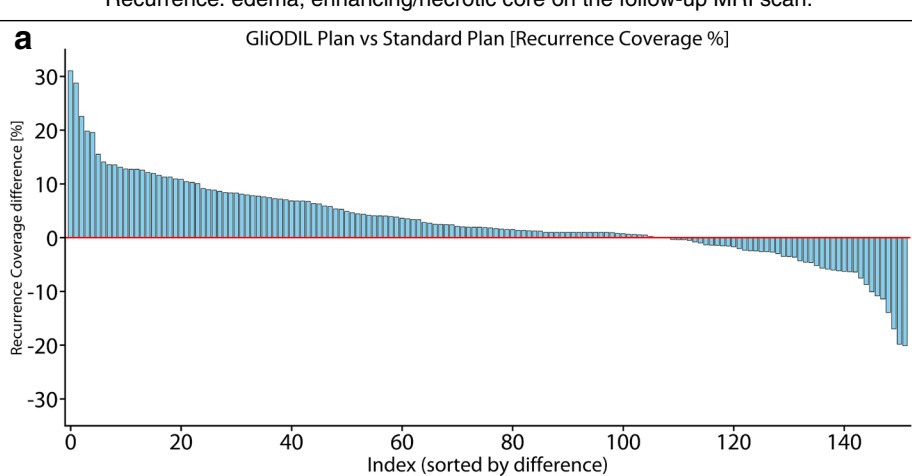

When GliODIL fails. Two bad cases.

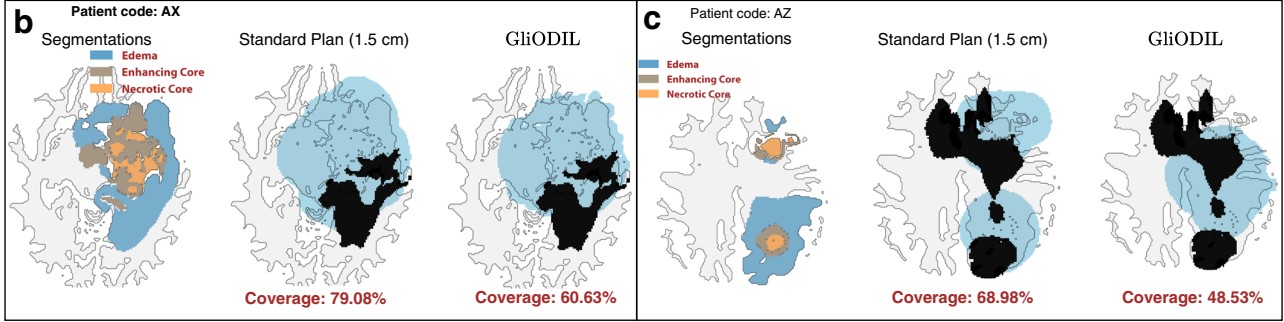

**Fig. 7 | Case-by-case comparison of real patient data and GliODIL limitations. a** Comparison of GliODIL with the Standard Plan across individual cases. **b, c** Examples of cases where GliODIL underperforms. Source data are provided with this paper.

was calibrated. A more sophisticated initial condition setup could potentially enhance GliODIL's handling of such rare complex cases.

Ending this section, we note that the $PDE_{GliODIL}$ model, through the comparative analysis detailed in section "Full Spatial Tumor Cell Distribution Inference", aligns with the $PDE_{CMA-ES}$ in capturing pre-operative data yet surpasses it in recurrence prediction accuracy. This suggests the parameter estimations by $PDE_{GliODIL}$ more accurately reflect patient-specific conditions. Such improvements imply the data-driven component of GliODIL benefits the fine-tuning of PDE parameters which can be used for both diagnostics and forcasting.

## Discussion

Addressing the unmet need for more adaptable tumor growth models, our GliODIL framework stands out by embodying essential features such as computational efficiency, error control through physics residual evaluation, and flexibility in model assumptions. These attributes enable GliODIL to adeptly navigate the complexities of tumor progression, even with limited initial data. Key experiments validate these capabilities, demonstrating GliODIL's superiority in predicting tumor recurrence and its significant advancements over traditional approaches, such as uniform margins and PDE solutions with calibrated parameters.

Leveraging its ability to synthesize information from diverse modalities, GliODIL significantly benefits from the integration of FET-PET imaging, which notably enhances its predictive accuracy beyond the capabilities of the Standard Radiotherapy Plan. This plan typically employs uniform safety margins around the visible tumor regions identified on MRI scans, aiming to delineate both the visible tumor and potential microscopic tumor extensions not visible on imaging. This

approach, while clinically practiced, lacks the specificity required for optimal therapeutic outcomes. The improvements GliODIL brings to the table, as evidenced by the patient outcomes depicted in Figs. 5 and 6, underscore the framework's superior capability to utilize complex imaging data to refine treatment strategies. Such performance enhancements are not achieved by models strictly adhering to PDE solutions, which struggle to capture the full extent of tumor recurrence, even with the addition of extra imaging inputs.

GliODIL's adaptive modeling approach provides a stark contrast to the Standard Radiotherapy Plan's reliance on uniform margins, offering a much-needed precision in targeting the tumor. GliODIL has the potential to be extended by integrating additional diagnostic modalities beyond FET-PET imaging, such as MR diffusion, perfusion imaging, and MRSI metabolic imaging. These modalities could provide deeper insights into tissue properties like cellularity, microstructure, and metabolism, enhancing the model's ability to represent complex tumor dynamics. Alternatively, using the same modalities, such as MRI and FET/PET, in a time-series format could provide significant benefits. Sequential imaging data would enable the calibration of more complex tumor growth models, which are otherwise limited by the ill-posed nature of single time-point data. This approach could offer a deeper understanding of tumor dynamics and improve model accuracy.

Our results thus highlight both the relevant contribution of advanced, biological imaging techniques to inform about the underlying tumor biology and the ability of our GliODIL framework to flexibly incorporate such additional information to improve inverse problem-solving validated by the clinically important tumor recurrence prediction task. Our current study is concentrated on immediate

post-operative treatment, necessitating reliance on a single imaging time point.

Moreover, addressing uncertainties in parameter inference remains critical, and future work could explore embedding variational inference techniques to enhance the framework's robustness.

The GliODIL framework, which utilizes multi-modal data and leverages PDEs for data-driven solution regularization to capture complex dynamics yet remains tunable with limited data, significantly outperforms models strictly governed by PDEs in forecasting tumor recurrence, as well as surpassing the uniform margin approaches that represent standard clinical practice. This underscores its considerable potential for solving diverse inverse problems in biology and highlights its promising prospects for widespread application. Moving forward, to advance research into tumor dynamics and the customization of treatment approaches, we provide access to a dataset that includes MRI images from 152 glioblastoma patients, 58 of whom have undergone pre-treatment FET-PET scans.

## Methods

The study was approved by the local ethics committee of the TU Munich (283/21 S-SR) and conducted in accordance with the criteria set by the Declaration of Helsinki. All patients provided written informed consent. All suitable patient data available at that time were used without exclusion. Furthermore, no potentially identifiable medical data are published: all segmentation maps, FET-PET maps and tissue maps have been anonymized.

### Tumor growth model

The core of our forward model rests on the Fisher-Kolmogorov Reaction-Diffusion (FK) equation, tailored for simulating tumor growth dynamics in terms of cellular diffusion and proliferation.

The partial differential equation (PDE) characterizing this model delineates the spatio-temporal evolution of the normalized tumor cell density $u(x, y, z, t)$ across a three-dimensional patient-specific brain anatomy segmented from MRI scans. The governing equation is:

$$\frac{\partial u(x,y,z,t)}{\partial t} = \nabla \cdot (D\nabla u) + \rho u(1 - u) \tag{1}$$

The proliferation rate of the tumor is denoted by $\rho$, while $D(x, y, z)$ serves as the spatially varying diffusion coefficient that captures the tumor's invasive characteristics. In the simulation, we enforce a no-flux boundary condition at the edges of the computational domain, confined to brain tissue.

We impose an initial condition in accordance with Eq. (1) as follows:

$$u(x, y, z, 0) = G(x, y, z) \tag{2}$$

where for $G(x, y, z)$ we employ a Gaussian function centered at an origin point $\vec{\mathbf{x}}_0$, as shown in Eq. (3):

$$G(\vec{\mathbf{x}}) = C_1 \exp\left(-\frac{(\vec{\mathbf{x}} - \vec{\mathbf{x}}_0)^2}{C_2}\right) \tag{3}$$

where we set the constants to $C_1 = \frac{50}{(60\pi)^{3/2}}$, $C_2 = 60$, which correspond to the initial tumor sizes depicted in Figs. 1 and 2.

For further definitions we assume discretization of the domain for computational purposes. Each voxel at coordinates $(i_x, i_y, i_z)$ is attributed a diffusion coefficient $D_{i,j,k}$, calculated as:

$$D_{i,j,k} = w_{i,j,k} D_w + g_{i,j,k} D_g \tag{4}$$

Here, $w_{i,j,k}$ and $g_{i,j,k}$ signify the proportions of white matter (WM) and gray matter (GM) at voxel $(i_x, i_y, i_z)$, respectively. $D_w$ and $D_g$ represent the diffusion coefficients associated with WM and GM. We make the assumption $D_w = R \cdot D_g$, with $R > 1$ being an unknown constant.

The residuals from Eqs. (1) and (2) are utilized to construct the loss function components $L_{PDE}$ and $L_{IC}$, respectively. These components are meant to quantify the divergence of the proposed tumor cell density $u(x, y, z, t)$ from the outcomes of the FK model. The process of discretizing $L_{IC}$ is straightforward, while the discretization approach for $L_{PDE}$ is delineated in section "Optimizing a Discrete Loss (ODIL)".

In this configuration, both the tumor's origin point $\vec{\mathbf{x}}_0 = (x_0, y_0, z_0)$ and the parameters associated with tumor dynamics $D, \rho, R$ are treated as unknowns that needs to be inferred.

### Optimizing a Discrete Loss (ODIL)

ODIL is a framework that addresses the challenges of solving inverse problems. It works by discretizing the PDE of the forward problem and using machine learning tools like automatic differentiation and popular deep learning optimizers (ADAM/L-BFGS) to minimize its residual while maintaining its sparse structure.

The previously introduced FK PDEs are discretized to perform numerical computations. We define $\Omega_1$ as the region within the brain where tumor cells can diffuse, primarily within the gray and white matter. Let's introduce the diffusion term $A$ and the reaction term $B$:

$$
\begin{aligned}
A\left(u_{i,j,k}^n\right) = & \frac{1}{\Delta x^2}\left(D_{i+\frac{1}{2},j,k}\left(u_{i+1,j,k}^n - u_{i,j,k}^n\right) - D_{i-\frac{1}{2},j,k}\left(u_{i,j,k}^n - u_{i-1,j,k}^n\right)\right) \\
& + \frac{1}{\Delta y^2}\left(D_{i,j+\frac{1}{2},k}\left(u_{i,j+1,k}^n - u_{i,j,k}^n\right) - D_{i,j-\frac{1}{2},k}\left(u_{i,j,k}^n - u_{i,j-1,k}^n\right)\right) \\
& + \frac{1}{\Delta z^2}\left(D_{i,j,k+\frac{1}{2}}\left(u_{i,j,k+1}^n - u_{i,j,k}^n\right) - D_{i,j,k-\frac{1}{2}}\left(u_{i,j,k}^n - u_{i,j,k-1}^n\right)\right)
\end{aligned}
\tag{5}
$$

$$B\left(u_{i,j,k}^n\right) = \rho u_{i,j,k}^n \left(1 - u_{i,j,k}^n\right) \tag{6}$$

Utilizing the Crank-Nicolson scheme, the residual loss $K$ of the PDE can be expressed as:

$$K_{i,j,k}^n = \frac{u_{i,j,k}^{n+1} - u_{i,j,k}^n}{\Delta t} - \frac{A\left(u_{i,j,k}^{n+1}\right) + A\left(u_{i,j,k}^n\right)}{2} - \frac{B\left(u_{i,j,k}^{n+1}\right) + B\left(u_{i,j,k}^n\right)}{2} \tag{7}$$

$$L_{PDE}(u, D_w, \rho, R) = \sum_{(i,j,k,n) \in \Omega_1} \left(K_{i,j,k}^n\right)^2 \tag{8}$$

Boundary conditions, particularly no-flux conditions outside $\Omega_1$, are employed to provide an accurate description of tumor cell behavior. The diffusion coefficients between gridpoints and within the tissue region $\Omega_1$ are computed as the average of their neighboring values.

The multi-grid ODIL technique, introduced in the paper[31], builds upon the original ODIL methodology by incorporating a multigrid decomposition scheme to fasten the convergence process. This technique is particularly adept at leveraging the multi-scale attributes of the forward problem. It decomposes the problem into various scale bands, each characterized by different levels of detail. Formally, for a uniform grid with dimensions $N_1 = N$, a hierarchical sequence of coarser grids is introduced with sizes defined as $N_i = N/2^{i-1}$ for $i = 1, \ldots, L$.

$$M_L(u_1, \ldots, u_L) = u_1 + T_1 u_2 + T_1 T_2 \ldots T_{L-1} u_L, \tag{9}$$

where each $u_i$ is a field on grid $N_i$, and $T_i$ serves as an interpolation operator mapping from coarser grid $N_{i+1}$ to its finer counterpart $N_i$. The discrete field $u$ on an $N$-sized grid is thus decomposed as $u = M_L(u_1, \ldots, u_L)$.

As depicted in Figs. 1,3 which illustrates the multigrid domain, a tumor growth is simulated over an ensemble of Cartesian 4D grids in

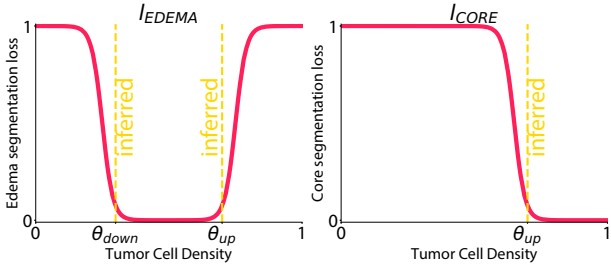

**Fig. 8 | Visualization of the segmentation penalty function with adaptive thresholds $\theta_{down}$ and $\theta_{up}$.** These thresholds are determined during the optimization process to adapt to the unique characteristics and sensitivities of different MRI images and tumor types.

both time and space, with each grid level being coarser than the preceding one. This hierarchical decomposition allows the optimization algorithm to initially concentrate on the coarse-scale features, incrementally incorporating finer-scale details as the process evolves. Such an approach not only enables a more comprehensive exploration of the parameter space but also sidesteps the pitfalls of local minima and expedites convergence.

We focus the computational grids on the tumor region. This leads to an average resolution of $71 \times 68 \times 55$ for the area of interest in our images, significantly reduced from the original $240 \times 240 \times 155$. For parameter inference within the GliODIL framework, we employed a $48^3$ spatial resolution and 192 degrees in the temporal resolution. Our tests indicated that this resolution is sufficient, revealing no significant differences compared to inference using the native resolution. For a forward PDE run with inferred parameters by GliODIL, which we call $PDE_{GliODIL}$, we use the native resolution.

### Imaging model

The imaging model we present serves as a bridge between the simulated tumor cell densities and the imaging signatures captured in MRI and FET-PET scans. This model translates tumor cell density, denoted as $u$, into quantifiable imaging signals that reflect observed clinical phenomena and imaging physics principles.

The core of the model's data-driven nature is encapsulated by the loss function $L_{DATA}$, which associates the simulated outputs with key imaging traits such as the tumor core, surrounding edema, and metabolic activity detected through FET-PET imaging. This loss function integrates four adjustable parameters $\{\theta_{down}, \theta_{up}, \theta_{PET}, \theta_{BKG}\}$ and is expressed as:

$$L_{DATA} = \lambda_{CORE}L_{CORE}(u, \theta_{up}) + \lambda_{EDEMA}L_{EDEMA}(u, \theta_{down}, \theta_{up}) + \lambda_{PET}L_{PET}(u, s, \theta_{BKG}) \tag{10}$$

where $\lambda_{CORE}, \lambda_{EDEMA}, \lambda_{PET}$ are weights. The loss function is composed of individual terms corresponding to distinct anatomical features:

- $L_{CORE}$ relates tumor cell concentrations above the threshold $\theta_{up}$ to the tumor core region
- $L_{EDEMA}$ delineates the edema area surrounding the tumor, regulated by the lower and upper thresholds $\theta_{down}$ and $\theta_{up}$.
- $L_{PET}$ reflects the metabolic activity as indicated by FET-PET signals, influenced by a scaling factor $\theta_{PET}$ and an offset $\theta_{BKG}$.

These adaptive parameters $\{\theta_{down}, \theta_{up}, \theta_{PET}, \theta_{BKG}\}$ enable the model to accommodate variations in MRI/FET-PET imaging contrasts and noise levels.

We adopt sigmoid functions to portray the gradational transitions observed at tumor region margins. The sigmoid, $\sigma(x)$, is specified as:

$$\sigma(x) = \frac{1}{1 + e^{-\beta x}} \tag{11}$$

Here, $\beta$ modulates the steepness of the transition and is set to $\beta = 50$. For the tumor core:

$$L_{CORE}(u, \theta_{up}) = \sum_{(i,j,k,n) \in \Omega_2} \sigma\left(\theta_{up} - u_{i,j,k}^n - \alpha\right) \tag{12}$$

for the edema:

$$L_{EDEMA}(u, \theta_{down}, \theta_{up}) = \sum_{(i,j,k,n) \in \Omega_2} \sigma\left(\theta_{down} - u_{i,j,k}^n - \alpha\right) + \left(1 - \sigma\left(\theta_{up} - u_{i,j,k}^n + \alpha\right)\right) \tag{13}$$

where $\alpha$ offsets the thresholds and is set to $\alpha = 0.05$.

In this context, $\Omega_2$ represents the collection of voxels that map to the time point at which the imaging is conducted; for single-image analysis, this corresponds to the final time slice. See Fig. 8 for the shape of segmentation penalty function.

The metabolic activity within the tumor is evaluated by the loss term $L_{PET}$, which aligns the simulated metabolic signal with actual FET-PET scan observations:

$$L_{PET}(u, \theta_{PET}, \theta_{BKG}) = \sum_{(i,j,k,n) \in \Omega_3} \left(u_{i,j,k}^n - p_{i,j,k}^{PET}\right)^2 \tag{14}$$

For this purpose, $\Omega_3$ is defined as the subset of $\Omega_2$ where voxels are attributed to the metabolically active regions, specifically the enhancing tumor core and the edema, as visualized in Fig. 1 in the feature extraction. Regions manifesting necrosis are omitted due to their lack of metabolic activity, and areas beyond the edema and enhancing core are also excluded to minimize noise interference, which is assumed to offer negligible informative value.

Here $p_{i,j,k}^{PET}$ is the normalized to $[0, 1]$ FET-PET signal $y_{i,j,k}^{PET}$ scaled by $\theta_{PET}$ and with an offset $\theta_{BKG}$:

$$p_{i,j,k}^{PET} = \theta_{PET} \cdot \left(y_{i,j,k}^{PET} - \theta_{BKG}\right) \tag{15}$$

The devised loss function $L_{DATA}$ quantifies the discrepancies between simulated tumor cell densities and empirical imaging data.

### Final loss function

The final loss function measures discrepancy between proposed tumor cell concentrations $u(x, y, z, t)$ and our objective. It is a composite term that comprises contributions from multiple sources: the PDE constraint, data fitting, and additional regularization term. Specifically, the PDE loss, denoted by $L_{PDE}$ (introduced in Optimizing a Discrete Loss (ODIL)), imposes the discretized PDE equation constraint. Initial condition loss term $L_{IC}$ in the overall loss function serves to enforce that the tumor at $t = 0$ originates from a Gaussian origin (introduced in Tumor Growth Model). The data loss (introduced in Imaging Model), denoted by $L_{DATA} = \lambda_{CORE}L_{CORE} + \lambda_{EDEMA}L_{EDEMA} + \lambda_{PET}L_{PET}$, accounts for matching tumor core and edema segmentations as well as fitting to PET metabolic data. Additional regularization term confines the inferred parameters within a plausible range $L_{PARAMS}$[35].

The overall loss function can thus be expressed as:

$$\mathcal{L} = \lambda_{PDE}L_{PDE} + \lambda_{IC}L_{IC} + \lambda_{CORE}L_{CORE} \tag{16}$$

**Table 1 | Parameter ranges for generating synthetic single-focal and multi-focal tumor datasets**

| Shared Parameters | | |
|---|---|---|
| **Parameter** | **Min** | **Max** |
| $D_w$ | 0.035 | 0.2 |
| $\rho$ | 0.035 | 0.2 |
| $R$ | 10 | 30 |
| $\theta_{necro}$ | 0.70 | 0.85 |
| $\theta_{up}$ | 0.45 | 0.60 |
| $\theta_{down}$ | 0.15 | 0.35 |
| $T_{sim}$ | 100 | — |
| Single Focal Tumor Center (mm) | | |
| $(x_0, y_0, z_0)$ | 57.6 | 96 |
| Multi-Focal Tumor Centers (mm) | | |
| Tumor 1 Center $(x_0^1, y_0^1, z_0^1)$ | 57.6 | 96 |
| Tumor 2 Center $(x_0^2, y_0^2, z_0^2)$ | $(x_0^1, y_0^1, z_0^1) \pm 9.6$ | |
| Tumor 3 Center $(x_0^3, y_0^3, z_0^3)$ | $(x_0^2, y_0^2, z_0^2) \pm 9.6$ | |

All values are given as minimum and maximum.

$$+ \lambda_{EDEMA} L_{EDEMA} + \lambda_{PET} L_{PET} + \lambda_{PARAMS} L_{PARAMS} \quad (17)$$

where $\lambda_*$ are detailed on in section "Full Spatial Tumor Cell Distribution Inference" experiments.

### Evaluation metrics

We introduce specific metrics to evaluate the performance of the proposed GliODIL framework. These metrics include the Dice score, the FET-PET signal correlation, and the Recurrence Coverage.

**Dice score.** The Dice score, also known as the Sørensen-Dice index or Dice coefficient, is a widely recognized metric in medical image analysis for quantifying the similarity between two volumes. The coefficient is defined as twice the area of overlap between the two volumes divided by the total number of voxels in both samples:

$$\text{Dice Coefficient} = \frac{2 \times |A \cap B|}{|A| + |B|} \quad (18)$$

where $A$ represents the thresholded tumor volume from a computational model and $B$ represents the segmented tumor volume from patient MRI segmentations or thresholded ground truth tumor volume. This score ranges from 0 to 1, where 0 indicates no overlap and 1 indicates perfect agreement between the two segmented regions.

**FET-PET signal correlation.** This metric calculates the Pearson's correlation coefficient between FET-PET signal intensity and tumor cell concentration in regions where a high degree of correlation is expected: the enhancing core and the edema.

**Recurrence coverage.** The principal metric for evaluating our model's effectiveness in radiotherapy planning is its accuracy in predicting tumor recurrence within the specified radiation volume after treatment. We identify the area of tumor recurrence using two definitions: a narrow definition focusing on the enhancing core observed in post-treatment patient data, and a broader definition that includes edema and the enhancing/necrotic core in the post-treatment follow-up MRI scans. We register the post-treatment patient anatomy to the pre-treatment to eliminate spatial shifts caused by surgeries. The Recurrence Coverage [%] metric refers to the percentage of segmented recurrences, that are encompassed within the radiation volume delineated by a given radiotherapy plan. Each radiotherapy plan maintains a uniform total volume, equivalent to that of the Standard Plan (CTV), which applies a consistent 1.5 cm margin around the pre-operative segmented enhancing and necrotic regions. To facilitate a fair comparison, we adjust the Standard Plan's total volume by excluding portions that extend beyond the patient's anatomical boundaries before making any calculations.

### Baselines

We compare our results with the Covariance Matrix Adaptation Evolution Strategy (CMA-ES) method[36,37], which relies on numerous simulations and employs a loss function from[21] to determine parameters that best fit the data. Its forward finite-difference scheme is analogous to that described in[38]. In addition, we compare with the Learn-Morph-Infer (LMI) technique[28], a deep learning framework for tumor growth model parameter estimation. We also include nnU-Net[39], the state-of-the-art neural network architecture for medical image segmentation, as an end-to-end data-driven baseline. For the standard nnU-Net experiments, the raw input features consisted of brain tissue distribution maps and pre-operative tumor segmentations. In contrast, for the nnU-Net-FET-PET variant, we additionally included FET-PET metabolic maps as input. The network outputs binary segmentation maps indicating predicted recurrence regions. To ensure compatibility with our radiotherapy planning pipeline and to standardize the output format with other models, we converted these binary outputs into a continuous representation using a Euclidean distance map. This transformation assigns a value of 1 to predicted recurrence regions, with values smoothly decreasing toward 0 as the distance from the predicted areas increases, resembling tumor cells distributionx. To maintain consistency between the GliODIL framework and nnU-Net results, we applied the same preprocessing steps across both methods. This included intensity normalization of input images, resampling to a common resolution, and brain mask application. The network was trained using an 80/20 train-test split. LMI employs data-driven convolutional neural networks to discover unknown PDE parameters, and CMA-ES is a state-of-the-art method[21] using both MRI and FET-PET data. We transitioned from TMCMC (Transitional Markov Chain Monte Carlo) to CMA-ES (Covariance Matrix Adaptation Evolution Strategy) to better scale our implementation for multiple patients. For clarity, solutions using the forward PDE finite difference method with parameters from CMA-ES and LMI are labeled as PDE$_{CMA-ES}$ and PDE$_{LMI}$, respectively, indicating direct PDE solutions.

### Synthetic dataset

For loss function weights calibration purposes, we created the synthetic dataset for single focal and localized multi-focal tumors by solving the system of PDEs using a traditional FDM solver. We use a tissue atlas to describe the spatial distribution of the brain tissues. We variate ground truth tumor growth model parameters, imaging model parameters and focal locations using uniform random distributions. In addition to the parameters outlined in Table 1, here we introduce an extra threshold, $\theta_{necro}$, above which the region is treated as a necrotic core without FET-PET metabolic activity. The range of parameters utilized in the generation is summarized in Table 1. In the creation of synthetic FET-PET images, we implement a sequence of processing steps to emulate real-world FET-PET imaging characteristics. Initially, we introduce spatially correlated noise using the Gibbs method to simulate the inherent noise in FET-PET scans. Following this, we remove the necrotic core area from the images, reflecting the typical absence of metabolic activity in these regions in actual FET-PET scans. Subsequently, we apply a downsampling process by a factor of 4, followed by an upsampling using zeroth-order spline interpolation. This sequence of downscaling and upscaling to an effective low resolution of 4mm is designed to simulate the lower resolution and partial volume effects commonly observed in real FET-PET images.

**Table 2 | Demographic and clinical characteristics of the study cohort**

| Characteristic | Value |
|---|---|
| Number of patients, $n$ | 152 |
| Age, years (mean ± SD) | 62.4 ± 10.8 |
| Diagnosis | WHO-CNS grade 4 glioblastoma |
|  | (IDH wild-type) |
| Deceased, $n$ (%) | 83 (54.6%) |
| Time to death, days (mean ± SD) | 467.7 ± 260.8 |

### Clinical dataset

We selected 152 adult patients from the glioma database at TUM University Hospital to validate our method in actual patient data. All patients were diagnosed with a WHO-CNS grade 4 IDH wild type glioblastoma, according to the 2021 WHO classification of brain tumors. Demographic and clinical characteristics of the study cohort are summarized in the Table 2. In the preoperative images, patients' tumor segmentations had average volumes of 18.5 cm$^3$ for the enhancing core, 58.9 cm$^3$ for edema, and 12.2 cm$^3$ for the necrotic core. In the postoperative follow-up, the average volumes decreased to 6.0 cm$^3$ for the enhancing core, 23.8 cm$^3$ for edema, and 6.4 cm$^3$ for the necrotic core. Imaging data included a preoperative and postoperative MRI, as well as an MRI scan at first tumor recurrence following combined radio-chemotherapy according to the Stupp protocol. MR scans were performed on a 3T Philips MRI scanner (either Achieva or Ingenia; Philips Healthcare, Best, The Netherlands) and comprised 3D-T1w-MPRAGE images before and after administration of Gadolinium-based contrast agent, 3D-FLAIR images and 3D-T2w images (all 1mm isotropic voxel resolution). In addition, for 58 patients, preoperative FET-PET imaging was also available. FET-PET data were acquired on either a PET/MR (Biograph mMR, Siemens Healthcare GmbH, Erlangen, Germany) or a PET/CT (Biograph mCT; Siemens Healthcare, Knoxville, TN, USA), according to a standard clinical protocol. Patients were asked to fast for a minimum of 4 h before scanning. Emission scans were acquired at 30–40 min after intravenous injection of a target dose of 185 ± 10 MBq [18F]-FET. Attenuation correction was performed according to vendor's protocol. We preprocess MRI and FET-PET images using BraTS Toolkit[32] resulting in images resolution of $240 \times 240 \times 155$ with segmented tumor regions. Given our assumption that surrounding tissues remain static, we segment brain tissues based on an atlas registration[21].

To compare GliODIL-derived radiotherapy plans (clinical target volume, CTV) with current standard-of-care plans, we followed the ESTRO-EANO guidelines[5] to generate standard CTV maps. In brief, we dilated the preoperative tumor segmentation (tumor core and contrast-enhancing tumor) by a uniform margin of 15mm, excluding non-brain and CSF areas from the target volume.

### Initial guess

We aim to solve the optimization problem for the model parameters referenced in a table in Fig. 1 as well the tumor concentration field $u(x, y, z, t)$ on a 4D discrete grid. A meaningful initial guess for these values is crucial for the time of the optimization process and the overall success. We assume the initial tumor location coordinates $x_0, y_0, z_0$ to be in the center of the tumor core. In addition, for the initial guess we assume $\{R, \theta_{PET}, \theta_{BKG}, \theta_{down}, \theta_{up}\}$ to be in the middle of the plausible range[35]. Here, we describe the procedure followed to obtain the remaining $\{u, D, \rho\}$:

1. Initiate a forward run propagation using characteristic values: $D_{ch} = \frac{V_{EDEMA}}{V_{CORE}}, \rho_{ch} = 1$. Here, $V_{EDEMA}$ and $V_{CORE}$ refer to the volumes of the edema and tumor core segmentations, respectively. Concurrently, track the Dice coefficients for both the tumor core and edema.

2. Terminate the forward run when a local maximum is reached for the segmentation volume-weighted sum of the Dice scores. Document the time at this instance as $T_{ch}$ and the tumor cell concentration as $u_{ch}$.

3. For the initial guess, we use $u = u_{ch}$ as the tumor cell concentration and $D = \frac{D_{ch}}{T_{ch}}, \rho = \frac{\rho_{ch}}{T_{ch}}$ as the initial dynamics.

For a comparative analysis between the initial guess and the PDE$_{GliODIL}$ results, see Fig. 2.

We also investigate how the initial guess affects solution quality by varying $D_{ch}, \rho_{ch}, x_0, y_0$, and $z_0$ by ± 20% using a uniform distribution. For real patient data (patient AA), the Core Dice coefficient was 0.81 ± 0.02, edema Dice 0.35 ± 0.05, and PET core Dice 0.65 ± 0.05, all achieved without changes in convergence time. These results indicate that reasonable variations in the initial guess only modestly affect segmentation fit quality. However, the initial guess remains important, as lacking it or using overly simplistic guesses (e.g., duplicating the initial condition across temporal resolution) can prevent the model from converging to meaningful solutions. Nonetheless, as long as the initial guess is reasonable, variations make little difference.

### Execution times and spatial resolutions

The execution times of the studied methods are summarized as follows. The model PDE$_{CMA\_ES}$ exhibited an evaluation time of 80 ± 55 min per patient. For GliODIL and its associated PDE$_{GliODIL}$, the patient evaluation time was significantly lower, at 45 ± 20 min, while nnU-Net was even faster, taking only a few seconds. It's important to note that the evaluations for GliODIL, PDE$_{GliODIL}$, and nnU-Net were conducted on a single GPU, whereas the computations for PDE$_{CMA\_ES}$ and PDE$_{LMI}$ were performed on 16 CPU cores.

In terms of spatial resolution, PDE$_{CMA\_ES}$ employs a forward Euler finite difference scheme with a progressively increasing spatial resolution, reaching a maximum of $96^3$ for the entire space. PDE$_{GliODIL}$, on the other hand, uses a Crank-Nicolson scheme with a resolution of $48^3$ for the tumor region, resulting in a comparable total spatial coverage. However, since the PDE in use is nonlinear, there are no guarantees regarding scaling the temporal resolution needed for stability at higher spatial resolutions. In contrast, PDE$_{LMI}$ operates on a fixed $128^3$ resolution, consistent with its training scale.

### Reporting summary

Further information on research design is available in the Nature Portfolio Reporting Summary linked to this article.

## Data availability

The patient data utilized (including all tissues, FET-PET scans, and segmentations) are available for reproduction and benchmarking at https://huggingface.co/datasets/m1balcerak/GliODIL. The recurrence source data accompanying this paper is available at https://github.com/m1balcerak/GliODIL[40]. Source data are provided with this paper.

## Code availability

The code for the proposed method, as well as for generating synthetic patients, can be found at https://github.com/m1balcerak/GliODIL[40].

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

## Acknowledgements

We extend our gratitude to Bastian Wittmann (University of Zurich) for his assistance with the manuscript. We also thank the Dagstuhl Seminar (2023) for fostering discussions that inspired this work. This research was supported by the Helmut Horten Foundation (M.B., B.M.), the European Cooperation in Science and Technology (COST) (B.M.), and by the National Science Foundation (NSF), Division of Mathematical Sciences (DMS) (J.L.), under grants NSF-DMS-1763272/Simons Foundation (QN594598, J.L.) and NSF-DMS-2309800 (J.L.).

## Author contributions

M.B. was responsible for the methodology, software implementation, and writing the original draft. J.W. contributed to the methodology and the implementation of baselines. P.K. handled the methodology and software implementation. I.E. was involved in the methodology. S.L.

worked on software implementation and the implementation of base-lines. P.K. contributed to the conceptualization and writing (review and editing). R.Z. and J.S.L. both were responsible for the methodology and writing (review). T.A. handled the implementation of baselines. I.Y. handled data acquisition. B.W. contributed to the methodology, writing (review and editing), and data acquisition. B.M. provided supervision, conceptualization, methodology, and writing (review and editing). B.W. and B.M. contributed equally as senior authors.

## Competing interests

The authors declare no competing interests.
