## [Transparent Peer Review file · Nature Communications]

Individualizing Glioma Radiotherapy Planning by Optimization of a Data and Physics-Informed Discrete Loss

Corresponding Author: Mr Michal Balcerak

Version 0:

Reviewer comments:

Reviewer #1

(Remarks to the Author)

I found this paper very interesting but I have some comments.

Why do you directly mention PET as a way to obtain more information about each tumor, without mentioning that more advanced MRI sequences can also be used to better understand the tumor infiltration? PET requires radioactive tracers and either a PET/MR scanner or a separate PET scan. For example, quantitative MRI (also called relaxometry) is becoming clinically available at more and more MR scanners, and quantitative T1- T2- and PD (proton density) values can be used to more easily detect tumor tissue (since the T1-, T2- and PD-values are then not within the normal range).

Nunez-Gonzalez, L., van Garderen, K. A., Smits, M., Jaspers, J., Romero, A. M., Poot, D. H., & Hernandez-Tamames, J. A. (2022). Pre-contrast MAGiC in treated gliomas: a pilot study of quantitative MRI. *Scientific Reports*, 12(1), 21820.

Blystad, I., Warntjes, J. M., Smedby, Ö., Lundberg, P., Larsson, E. M., & Tisell, A. (2017). Quantitative MRI for analysis of peritumoral edema in malignant gliomas. *PLoS One*, 12(5), e0177135.

Bahrami, N., Hartman, S. J., Chang, Y. H., Delfanti, R., White, N. S., Karunamuni, R., ... & McDonald, C. R. (2018). Molecular classification of patients with grade II/III glioma using quantitative MRI characteristics. *Journal of neuro-oncology*, 139, 633-642.

Furthermore, MRI sequences for studying micro diffusion are also sensitive to tumor infiltration. Standard diffusion results in metrics like fractional anisotropy (FA), while micro diffusion gives micro FA which is more sensitive.

Szczepankiewicz, F., Lasič, S., van Westen, D., Sundgren, P. C., Englund, E., Westin, C. F., ... & Nilsson, M. (2015). Quantification of microscopic diffusion anisotropy disentangles effects of orientation dispersion from microstructure: applications in healthy volunteers and in brain tumors. *Neuroimage*, 104, 241-252.

For radiotherapy treatment planning, do you use manual planning or an automatic planner to obtain the optimized radiotherapy plans? If you use an automatic planner, which one? Was the time for generating the treatment plans similar for MRI and MRI + PET? How can the PET information be used by an automatic planner? Or are your plans here simply the tumor segmentations + margin? From my perspective a radiotherapy plan means the actual dose (measured in Gray) to deliver in each voxel, and how to achieve that using a number of small and large radiation shots in different locations. To create the plan has historically been done manually (time consuming), but today different automatic planners are available. You should make it clear that you don't show any plans, but that you show segmentations if my understanding is correct.

In treatment planning you normally talk about GTV, CTV, PTV; gross tumor volume, clinical target volume, planning target volume, where CTV is larger than GTV and PTV is larger than CTV. You normally don't talk so much about enhancing core

and edema. PTV can for example be CTV + margin of 3-4 mm. It would be good if you can add GTV CTV and PTV to your paper, and explain if your model is supposed to replace standard CTV or standard PTV or both. Also explain if "radiotherapy volume" is equivalent to PTV or to something else.

Did you export your data from the oncology department, or did you only export MR images from the radiology department? Data from the oncology department should have GTV CTV PTV segmentations (normally done manually), and also segmentations of different organs at risk (OAR) to be avoided. It is these clinical segmentations, used in the clinical radiotherapy oncology workflow, that should be used, stored as DICOM RTSTRUCT, rather than automatic segmentations from BraTS toolkit. The data from the oncology department also has the dose distribution volume, stored as DICOM RTDOSE, and the plan used stored as DICOM RTPLAN.

Were all treatment plans for the same level of radiation (e.g. 60 Gray), or how much does the prescribed dose vary across patients? Can you see any pattern regarding if your approach works better or worse for higher or lower radiation doses? You should also mention the tumor sizes (at least min max median) for your patients.

How / where are you sharing the dataset? It should be stated in the paper.

How does your model compare to the model used in this work?

Jaroudi, R., Åström, F., Johansson, B. T., & Baravdish, G. (2020). Numerical simulations in 3-dimensions of reaction-diffusion models for brain tumour growth. *International Journal of Computer Mathematics*, 97(6), 1151-1169.

(Remarks on code availability)

Reviewer #2

(Remarks to the Author)

This paper introduces GliODIL, an optimization framework designed to personalize glioma radiotherapy planning by combining data-driven techniques with traditional numerical methods. The framework leverages multimodal imaging, including MRI and FET-PET, and integrates a Fisher-Kolmogorov type physics model (PDE) to describe tumor growth. By optimizing a discrete loss function, GliODIL aims to infer the full spatial distribution of tumor cell concentrations, enhancing the prediction of tumor recurrence and improving radiotherapy planning beyond traditional uniform margin approaches. The proposed framework was validated on both synthetic and real patient data, and compared with an evolutionary strategy optimization method, a deep-learning-based method, and standard clinical practice, where the authors claim significant improvements in computational efficiency and prediction accuracy. However, the paper still has areas that require improvement and caution:

1. The organization of this paper could be improved. It would be beneficial if the authors could further outline the complete workflow of the GliODIL. A detailed step-by-step description of the establishment of PDE-based loss, imaging-based loss, and how they interact with each other would greatly enhance understanding. Specifically, regarding the imaging component, this should include the integration points of MRI and FET-PET data, how these data are processed and regularize the optimization of the model.
2. The paper should provide more detailed descriptions of a large number of parameters involved in the PDE model and imaging model. For example:
 - Diffusion Coefficient (D): Explain how D varies spatially and how it is estimated or inferred from data. The authors model the diffusion coefficient solely on the quantification of white matter and gray matter (decomposed into DW and DG) but lack further explanation.
 - Adaptive Parameters: The four adaptive parameters θ_{down} , θ_{up} , θ_{PET} and θ_{BKG} , lack detailed explanations. The paper should describe the initial values, and how these parameters are set and associated with the MRI and PET data.
 - Weight coefficients (λ^*): There are several regularization terms involving in the imaging-based loss and overall loss whose balanced weight should be detailed.
3. The results section seems to lack a clear and logical structure. It would be beneficial to divide it into distinct subsections with clear headings such as "Synthetic Data Validation," "Real Patient Data Validation," "Comparison with Existing Methods," and "Clinical Relevance."
4. The authors should provide a more detailed explanation of the data used in the study, particularly the synthetic and real image data, and how the corresponding ground truth is established. This would enhance the clarity and understanding of the results. For example, include visual representations of the data and ground truth in the comparison figures (e.g., Figure 2 and Figure 3). Each figure should provide clear examples of the input data, the predicted results from the GliODIL framework, and the corresponding ground truth for direct comparison. Add panels that show the ground truth images alongside the model predictions. This would allow readers to visually assess the accuracy and quality of the model's performance. Additionally, the acquisition of ground truth for real-world data in this study remains unclear.
5. The reliance on solving the PDE at a single time point can indeed make the results heavily dependent on the initial guess, which can propagate errors and result in suboptimal predictions if the initial conditions are not accurate. The paper should have more discussion and experiments about the initial guess part.

6. The current comparison methods used in the paper are all based on PDEs, even when incorporating deep learning to estimate PDE parameters. Considering the availability of high-quality data, it is feasible to explore end-to-end deep learning approaches that directly predict the so-called invisible boundaries from MRI or PET images. If there is sufficient high-quality labeled data, is it possible to train deep learning models to predict the invisible boundaries of tumors directly from MRI or PET images?

7. Many assumptions in the paper, such as the initial tumor location and diffusion coefficients, might be overly idealized. The robustness and adaptability of the model under high uncertainty in initial conditions and parameters should be explored.

8. The dataset mainly consists of WHO-CNS grade 4 IDH wild-type glioblastoma patients. To verify the broad applicability of GliODIL, tests on different types and grades of gliomas should be conducted, and the results reported.

9. Model validation primarily relies on internal datasets, lacking independent external dataset validation. To enhance credibility and generalizability, GliODIL should be tested on independent external datasets, and those results should be reported.

10. The paper highlights the success of GliODIL but does not delve into cases where the model performs poorly. Detailed analysis of failure cases and the reasons behind these failures should be included to guide future improvements.

(Remarks on code availability)

Reviewer #3

(Remarks to the Author)
see attached

(Remarks on code availability)

Version 1:

Reviewer comments:

Reviewer #1

(Remarks to the Author)
The authors have addressed my comments.

(Remarks on code availability)

Reviewer #2

(Remarks to the Author)

The authors have made revisions in response to the reviewers' comments, including providing additional clarifications on the workflow, expanding details on parameter settings, reorganizing the Results section for better readability, which reflect their efforts to improve the manuscript based on the feedback. Some additional suggestions are listed below,

1. While the authors have added an overview of the workflow at the beginning of the Results section, I suggest avoiding the direct repetition of content from Section 4 (Methods). Instead, consider providing a concise summary of the methodology that directly references and aligns with Figure 1, which starts from the input to the output of the model. Additionally, I recommend reorganizing Figure 1 to enhance clarity and coherence. The current layout appears somewhat disorganized, making it challenging to follow the connections between the workflow components. Including clear annotations or labels that explain the equations and their relevance to specific steps within the workflow would further improve the figure's usability.

2. The optimal weight for λ_{PDE} (set to 1.0) was determined experimentally for the synthetic dataset. Would patient-specific optimization of this parameter improve the model's adaptability and performance?

3. In the comparison with nnUNET, could you provide more details about the experimental setup? Specifically, what were the raw input features used for the baseline nnUNET experiments, and how were the inputs modified or extended for the nnUNET-FETPET experiments? Additionally, were any specific preprocessing steps applied to ensure consistency between the GliODIL framework and nnUNET results?

(Remarks on code availability)

Reviewer #3

(Remarks to the Author)
Thank you for addressing my edits. Strong revision.

no further comments

(Remarks on code availability)

Version 2:

Reviewer comments:

Reviewer #2

(Remarks to the Author)

Thanks for the revision.

(Remarks on code availability)

Response to Reviewers

We sincerely thank all the reviewers for their valuable feedback and constructive suggestions on our manuscript titled "Individualizing Glioma Radiotherapy Planning by Optimization of a Data and Physics-Informed Discrete Loss" (GliODIL framework). Your insights have been instrumental in improving the quality and clarity of our work.

In response to your comments, we have undertaken a major revision of the manuscript. We have provided more detailed explanations of the radiotherapy plan definitions to align with standard clinical practices. Additionally, we have introduced a new baseline by incorporating an end-to-end deep learning approach, allowing for a comprehensive comparison with our physics-informed framework.

We have also included analyses that identify scenarios where our method may underperform, outlining these cases to guide future research directions. Furthermore, we have strengthened the justification for using FET-PET by elaborating on its advantages and relevance in our study, supported by relevant literature..

Moreover, we have provided detailed information about the patient cohort, including imaging protocols and dosages, survival times, and comprehensive patient characteristics. These additions ensure a thorough understanding of the clinical context.

We appreciate the opportunity to refine our manuscript and believe that the changes made significantly enhance its contribution to the field.

Thank you once again for your constructive feedback and support.

Sincerely, Authors

Reviewer #1 (Biomedical imaging, method development):

I found this paper very interesting but I have some comments.

Why do you directly mention PET as a way to obtain more information about each tumor, without mentioning that more advanced MRI sequences can also be used to better understand the tumor infiltration? PET requires radioactive tracers and either a PET/MR scanner or a separate PET scan. For example, quantitative MRI (also called relaxometry) is becoming clinically available at more and more MR scanners, and quantitative T1- T2- and PD (proton density) values can be used to more easily detect tumor tissue (since the T1-, T2- and PD-values are then not within the normal range).

Nunez-Gonzalez, L., van Garderen, K. A., Smits, M., Jaspers, J., Romero, A. M., Poot, D. H., & Hernandez-Tamames, J. A. (2022). Pre-contrast MAGiC in treated gliomas: a pilot study of quantitative MRI. *Scientific Reports*, 12(1), 21820.

Blystad, I., Warntjes, J. M., Smedby, Ö., Lundberg, P., Larsson, E. M., & Tisell, A. (2017). Quantitative MRI for analysis of peritumoral edema in malignant gliomas. *PLoS One*, 12(5), e0177135.

Bahrami, N., Hartman, S. J., Chang, Y. H., Delfanti, R., White, N. S., Karunamuni, R., ... & McDonald, C. R. (2018). Molecular classification of patients with grade II/III glioma using quantitative MRI characteristics. *Journal of neuro-oncology*, 139, 633-642.

Furthermore, MRI sequences for studying micro diffusion are also sensitive to tumor infiltration. Standard diffusion results in metrics like fractional anisotropy (FA), while micro diffusion gives micro FA which is more sensitive.

Szczepankiewicz, F., Lasič, S., van Westen, D., Sundgren, P. C., Englund, E., Westin, C. F., ... & Nilsson, M. (2015). Quantification of microscopic diffusion anisotropy disentangles effects of orientation dispersion from microstructure: applications in healthy volunteers and in brain tumors. *Neuroimage*, 104, 241-252.

Thank you for your appreciation of our work and this important comment. We agree that advanced MRI sequences, such as diffusion tensor imaging (DTI) or perfusion-weighted imaging, hold great promise to inform tumor growth models like GliODIL. Our choice of FET-PET was based on several factors: a) the well-documented correlation between FET uptake and tumor cell density, b) the established clinical role of FET-PET in gliomas (as exemplified in the recently published PET-RANO criteria), and c) the availability of FET-PET in our clinic through a strong collaboration between the Departments of Nuclear Medicine and Neuroradiology. Nonetheless, we acknowledge that incorporating advanced MRI modalities is an important future research direction, and we have added a short paragraph to the '3. Discussion' section reflecting this outlook.

ADDED TEXT:

GliODIL's adaptive modeling approach provides a stark contrast to the Standard Radiotherapy Plan's reliance on uniform margins, offering a much-needed precision in targeting the tumor. GliODIL has the potential to be extended by integrating additional diagnostic modalities beyond FET-PET imaging, such as MR diffusion, perfusion imaging, and MRSI metabolic imaging. These modalities could provide deeper insights into tissue properties like cellularity, microstructure, and metabolism, enhancing the model's ability to represent complex tumor dynamics. Alternatively, using the same modalities, such as MRI and FET/PET, in a time-series format could provide significant benefits. Sequential imaging data would enable the calibration of more complex tumor growth models, which are otherwise limited by the ill-posed nature of single time-point data. This approach could offer a deeper understanding of tumor dynamics and improve model accuracy.}

Our results thus highlight both the relevant contribution of advanced, biological imaging techniques to inform about the underlying tumor biology and the ability of our GliODIL framework to flexibly incorporate such additional information to improve inverse problem-solving validated by the clinically important tumor recurrence prediction task. Our current study is concentrated on immediate post-operative treatment, necessitating reliance on a single imaging time point.

For radiotherapy treatment planning, do you use manual planning or an automatic planner to obtain the optimized radiotherapy plans? If you use an automatic planner, which one? Was the time for generating the treatment plans similar for MRI and MRI + PET? How can the PET information be used by an automatic planner? Or are your plans here simply the tumor segmentations + margin? From my perspective a radiotherapy plan means the actual dose (measured in Gray) to deliver in each voxel, and how to achieve that using a number of small and large radiation shots in different locations. To create the plan has historically been done manually (time consuming), but today different automatic planners are available. You should make it clear that you don't show any plans, but that you show segmentations if my understanding is correct.

This is an important clarification. When discussing individual voxel-wise doses, one refers to the Planning Target Volume (PTV). In our work, we focus on the Clinical Target Volume (CTV). To be as close as possible to clinical application, our CTV definition follows the most recent ESTRO-EANO guidelines (Niyazi et al., Radiother Oncol 2023), which recommend a 15mm margin around the tumor core. This CTV is corrected for anatomical barriers, such as the skull or ventricles. In clinical practice, CTV is defined by a radiation oncologist and forms the basis for a medical physicist to calculate the PTV. To benchmark GliODIL against the clinical standard of care, we compared the CTV to volumes defined by GliODIL. We have edited our '2. Results' section to more clearly reflect this.

The MRI and MRI+PET scans have similar run times, approximately 30 minutes on our equipment. This is due to the parallelization of the GPU.

ADDED TEXT:

Clinical Standard and Its Limitations

In current clinical practice, radiotherapy planning for glioblastoma patients involves defining several target volumes to guide treatment. First, the Gross Tumor Volume (GTV) is delineated based on the visible tumor core identified in preoperative MRI scans. Next, a uniform margin—commonly around 1.5 centimeters \cite{Wen2010}—is added to the GTV to create the Clinical Target Volume (CTV), thereby accounting for potential microscopic tumor infiltration not visible on imaging. Finally, a Planning Target Volume

(PTV) is formed by expanding the CTV by an additional margin (often on the order of 3–4 millimeters) to compensate for setup uncertainties and patient movement during treatment.

However, this standardized process, especially the use of a fixed uniform margin to generate the CTV, does not reflect the highly heterogeneous nature of glioblastoma infiltration, variations in individual patient anatomy, or the presence of anatomical barriers that may influence tumor spread. As a result, the current approach may either over-treat healthy brain tissue—leading to unnecessary side effects—or fail to adequately cover infiltrative tumor regions that extend beyond the prescribed margin, thus increasing the risk of recurrence. These limitations highlight the need for more personalized and anatomically informed radiotherapy planning strategies.

[...]

We focus on improving the delineation of the CTV. Rather than simply applying a uniform margin to the GTV, our proposed method aims to replace the standard, margin-based CTV definition with a more patient-specific, model-driven approach.

In treatment planning you normally talk about GTV, CTV, PTV; gross tumor volume, clinical target volume, planning target volume, where CTV is larger than GTV and PTV is larger than CTV. You normally don't talk so much about enhancing core and edema. PTV can for example be CTV + margin of 3-4 mm. It would be good if you can add GTV CTV and PTV to your paper, and explain if your model is supposed to replace standard CTV or standard PTV or both. Also explain if "radiotherapy volume" is equivalent to PTV or to something else.

Please also see our answer to your comment above. We compare against (and aim to replace) CTV and better clarify this in the manuscript.

Did you export your data from the oncology department, or did you only export MR images from the radiology department? Data from the oncology department should have GTV CTV PTV segmentations (normally done manually), and also segmentations of different organs at risk (OAR) to be avoided. It is these clinical segmentations, used in the clinical radiotherapy oncology workflow, that should be used, stored as DICOM RTSTRUCT, rather than automatic segmentations from BraTS toolkit. The data from the oncology department also has the dose distribution volume, stored as DICOM RTDOSE, and the plan used stored as DICOM RTPLAN.

We indeed generated the CTV as described above from tumor and anatomy segmentations. Our hospital serves as a central neurosurgery hub for a large region; thus, many patients who get operated here, later receive radiotherapy closer to their place of residence and would have been lost to our study.

Were all treatment plans for the same level of radiation (e.g. 60 Gray), or how much does the prescribed dose vary across patients? Can you see any pattern regarding if your approach works better or worse for higher or lower radiation doses? You should also mention the tumor sizes (at least min max median) for your patients.

We report descriptive statistics about patients' and tumor characteristics in section '5.4 Clinical Dataset'. Since we generate and compare CTV, we did not directly infer individual, voxel-wise doses, but rather assumed a uniform distribution of dose (60 Gy) across our CTV (and GliODIL volume). We want to emphasize that GliODIL is meant to replace CTV estimation (based on which a PTV is then planned), and as such, we compare to the current clinical CTV generation algorithm (Niyazi et al., *Radiother Oncol* 2023). Our results demonstrate a clear superiority of the CTV based on GliODIL's tumor density isolines compared to the current clinical state-of-the-art. These findings provide a strong foundation and motivation for future clinical evaluation, where aspects such as PTV calculation will be important considerations. Nonetheless, our study presents compelling evidence that GliODIL offers a superior CTV estimation method.

ADDED TEXT:

Clinical Dataset

We selected 152 adult patients from the glioma database at TUM University Hospital to validate our method in actual patient data. All patients were diagnosed with a WHO-CNS grade 4 IDH wild type glioblastoma, according to the 2021 WHO classification of brain tumors. The average age was 62.4 ± 10.8 years, and among 83 deceased patients, the average time until death was 467.7 ± 260.8 days. In the preoperative images, patients' tumor segmentations had average volumes of 18.5 cm^3 for the enhancing core, 58.9 cm^3 for edema, and 12.2 cm^3 for the necrotic core. In the postoperative follow-up, the average volumes decreased to 6.0 cm^3 for the enhancing core, 23.8 cm^3 for edema, and 6.4 cm^3 for the necrotic core. Imaging data included a preoperative and postoperative MRI, as well as an MRI scan at first tumor recurrence following combined radio-chemotherapy according to the Stupp protocol. MR scans were performed on a 3T Philips MRI scanner (either Achieva or Ingenia; Philips Healthcare, Best, The Netherlands) and comprised 3D-T1w-MPRAGE images before and after administration of Gadolinium-based contrast agent, 3D-FLAIR images and 3D-T2w images (all 1mm isotropic voxel resolution). In addition, for 58 patients, preoperative FET-PET imaging was also available. FET-PET data were acquired on either a PET/MR (Biograph mMR, Siemens Healthcare GmbH, Erlangen, Germany) or a PET/CT (Biograph mCT; Siemens Healthcare, Knoxville, TN, USA), according to a standard clinical protocol. Patients were asked to fast for a minimum of 4 h before scanning. Emission scans were acquired at 30 to 40 min after intravenous injection of a target dose of $185 \pm 10 \text{ MBq}$ [^{18}F]-FET. Attenuation correction was performed according to vendor's protocol. We preprocess MRI and FET-PET images using BraTS Toolkit \cite{kofler2020brats} resulting in images resolution of $(240 \times 240 \times 155)$ with segmented tumor regions. Given our assumption that surrounding tissues

remain static, we segment brain tissues based on an atlas registration \cite{lipkova2019personalized}.

To compare GliODIL-derived radiotherapy plans (clinical target volume, CTV) with current standard-of-care plans, we followed the ESTRO-EANO guidelines \cite{Niyazi2023} to generate standard CTV maps. In brief, we dilated the preoperative tumor segmentation (tumor core and contrast-enhancing tumor) by a uniform margin of 15mm, excluding non-brain and CSF areas from the target volume.}

How / where are you sharing the dataset? It should be stated in the paper.

The dataset is available on the internet freely at <https://huggingface.co/datasets/m1balcerak/GliODIL> for anyone to download - with more than 120 downloads so far and counting. This information is in the Data and Code Availability section that points to our online resources (software code and data link).

ADDED TEXT:

The synthetic data generated for this study, along with the 152 real patient data utilized (FET-PET and segmentations), are both available for reproduction and benchmarking using the resources provided at github.com/m1balcerak/GliODIL/.

How does your model compare to the model used in this work?

Jaroudi, R., Åström, F., Johansson, B. T., & Baravdish, G. (2020). Numerical simulations in 3-dimensions of reaction–diffusion models for brain tumour growth. *International Journal of Computer Mathematics*, 97(6), 1151-1169.

The paper you mention describes a forward PDE Fisher-Kolmogorov model that uses finite difference as a numerical scheme. The setup (finite difference) is the exact same forward model as in PDE CMA-ES and is already included in the comparisons. They also use the same type of growth: logistic. The forward model alone needs a calibration method (to match the model with real data, i.e., medical scans in our case), which the authors did not provide. In our study, we employ CMA-ES for the model calibration. We highlight this connection in section '5.4 Baselines' where we cite Jaroudi et al.

ADDED TEXT:

We compare our results with the Covariance Matrix Adaptation Evolution Strategy (CMA-ES) method \cite{martin2021korali, weidner2024learnable}, which relies on numerous simulations and employs a loss function from \cite{lipkova2019personalized} to determine parameters that best fit the data. Its forward finite-difference scheme is analogous to that described in \cite{jaroudi2020numerical}.

Reviewer #2 (AI assisted radiotherapy imaging):

This paper introduces GliODIL, an optimization framework designed to personalize glioma radiotherapy planning by combining data-driven techniques with traditional numerical methods. The framework leverages multimodal imaging, including MRI and FET-PET, and integrates a Fisher-Kolmogorov type physics model (PDE) to describe tumor growth. By optimizing a discrete loss function, GliODIL aims to infer the full spatial distribution of tumor cell concentrations, enhancing the prediction of tumor recurrence and improving radiotherapy planning beyond traditional uniform margin approaches. The proposed framework was validated on both synthetic and real patient data, and compared with an evolutionary strategy optimization method, a deep-learning-based method, and standard clinical practice, where the authors claim significant improvements in computational efficiency and prediction accuracy. However, the paper still has areas that require improvement and caution:

1. The organization of this paper could be improved. It would be beneficial if the authors could further outline the complete workflow of the GliODIL. A detailed step-by-step description of the establishment of PDE-based loss, imaging-based loss, and how they interact with each other would greatly enhance understanding. Specifically, regarding the imaging component, this should include the integration points of MRI and FET-PET data, how these data are processed and regularize the optimization of the model.

Thank you for the positive feedback on our work. We provide a detailed outline of the method in Section 4 (Methods). Specifically, the PDE-based loss is established in Sections 4.1 and 4.2, the imaging-based loss is detailed in Section 4.3, and the final integrated loss, unifying the PDE- and imaging-based components, is explained in Section 4.4. To improve the readability of Section 2 (Results), we have also included a brief overview of the method and the key steps highlighted by the reviewer.

2. The paper should provide more detailed descriptions of a large number of parameters involved in the PDE model and imaging model. For example:

Diffusion Coefficient (D): Explain how D varies spatially and how it is estimated or inferred from data. The authors model the diffusion coefficient solely on the quantification of white matter and gray matter (decomposed into DW and DG) but lack further explanation.

Adaptive Parameters: The four adaptive parameters θ_{down} , θ_{up} , θ_{PET} and θ_{BKG} , lack detailed explanations. The paper should describe the initial values, and how these parameters are set and associated with the MRI and PET data.

Weight coefficients (λ^*): There are several regularization terms involving in the imaging-based loss and overall loss whose balanced weight should be detailed.

Both (a) Tumor Growth Parameters and (b) Imaging Model Parameters, as illustrated in Figure 1, are learned during the tumor cell inference process. These parameters are optimized by the Adam optimizer to minimize the loss function described in Section 4.4 (Final Loss Function). Regarding the 'adaptive parameters' mentioned by the reviewer, they are explained in detail in Section 4.3 (Imaging Model). These parameters are

adaptive, meaning they are not fixed to specific MRI/FET-PET instruments but are instead optimized within a plausible range, ensuring flexibility and applicability across different imaging setups.

3. The results section seems to lack a clear and logical structure. It would be beneficial to divide it into distinct subsections with clear headings such as "Synthetic Data Validation," "Real Patient Data Validation," "Comparison with Existing Methods," and "Clinical Relevance."

Thank you for the suggestion. We have revised the Results section to improve its organization and provide a clearer structure for readers, along with a more transparent explanation of the experimental flow and motivations. The subsections in the Results section are now reorganized as follows:

- a) Full Spatial Tumor Cell Distribution Inference
- b) Clinical Dataset Validation: Radiotherapy Planning and Recurrence Coverage

Additionally, the extended baseline explanation has been relocated to Section 5.2 (Baselines).

4. The authors should provide a more detailed explanation of the data used in the study, particularly the synthetic and real image data, and how the corresponding ground truth is established. This would enhance the clarity and understanding of the results. For example, include visual representations of the data and ground truth in the comparison figures (e.g., Figure 2 and Figure 3). Each figure should provide clear examples of the input data, the predicted results from the GliODIL framework, and the corresponding ground truth for direct comparison. Add panels that show the ground truth images alongside the model predictions. This would allow readers to visually assess the accuracy and quality of the model's performance. Additionally, the acquisition of ground truth for real-world data in this study remains unclear.

We have added a detailed description of the data processing and annotation in Section 5.4 (Clinical Dataset). Regarding Figures 2 and 3, these relate to synthetic data generated from a forward model run. While we primarily provide the segmentation/metabolic map, the ground truth is explicitly displayed in Figure 2b for clarity.

5. The reliance on solving the PDE at a single time point can indeed make the results heavily dependent on the initial guess, which can propagate errors and result in suboptimal predictions if the initial conditions are not accurate. The paper should have more discussion and experiments about the initial guess part.

Please refer to Section 5.5 (Initial Guess), where we provide detailed statistics on the assumptions made for the initial guess and analyze the variance in fit quality under different initial conditions.

6. The current comparison methods used in the paper are all based on PDEs, even when incorporating deep learning to estimate PDE parameters. Considering the availability of high-quality data, it is feasible to explore end-to-end deep learning approaches that directly predict the so-called invisible boundaries from MRI or PET images. If there is sufficient high-quality labeled data, is it possible to train deep learning models to predict the invisible boundaries of tumors directly from MRI or PET images?

Thank you for this insightful suggestion. We included a nnUNet-based deep learning pipeline as a comparison. This framework automatically configures all hyperparameters and is trained end-to-end to predict the recurrence region. Using raw imaging data as input, we trained nnUNets with the same 75/25 train/test split. The results are documented in Section 2, Results. However, it is important to note that this setup performs relatively poorly, emphasizing the necessity of a physics-informed framework given the current scarcity of high-quality labeled data.

7. Many assumptions in the paper, such as the initial tumor location and diffusion coefficients, might be overly idealized. The robustness and adaptability of the model under high uncertainty in initial conditions and parameters should be explored.

We kindly refer to our answer to the comment #5, where we explore GliODIL under varying initial guesses, meaning the Section 5.5 (Initial Guess).

8. The dataset mainly consists of WHO-CNS grade 4 IDH wild-type glioblastoma patients. To verify the broad applicability of GliODIL, tests on different types and grades of gliomas should be conducted, and the results reported.

Our focus on WHO-CNS grade 4 glioblastoma is deliberate, driven by two key factors: (a) the availability of glioblastoma cases with FET-PET and follow-up data, and (b) the goal of assessing the clinical value of GliODIL within a biologically well-defined entity without mixing tumor types. In Section 5.4 (Clinical Dataset), we provide detailed statistics on the sizes of pre-operative tumors and their recurrences.

9. Model validation primarily relies on internal datasets, lacking independent external dataset validation. To enhance credibility and generalizability, GliODIL should be tested on independent external datasets, and those results should be reported.

Thank you for your comment. We would like to clarify that, unlike deep learning frameworks, GliODIL does not require training data and operates solely at test time. Additionally, it works on segmentation maps, which significantly reduces data variability compared to deep learning models that process raw imaging data. The validation conducted within our cohort was performed on an unseen test set, as the hyperparameters were tuned using the synthetic dataset described in Section 5.3, highlighting the robust generalization capability of GliODIL. Our dataset, specifically curated for testing methods like ours, is the largest publicly available dataset of glioblastoma pre-op/follow-up cases. We aim to inspire researchers to collect more data and benchmark our method against their best approaches. Detailed protocols for obtaining hyperparameters, with fixed random seeds to ensure reproducibility, are provided. Additionally, the source code is openly available at <https://github.com/m1balcerak/GliODIL>.

10. The paper highlights the success of GliODIL but does not delve into cases where the model performs poorly. Detailed analysis of failure cases and the reasons behind these failures should be included to guide future improvements.

We appreciate this suggestion. A detailed investigation into failure cases is presented at the end of Section 2, Results. Please refer to Figure 7 and the accompanying text for the analysis of these cases.

Reviewer #3 (Mathematical Biology, cancer modeling):

Review: Individualizing Glioma Radiotherapy Planning by Optimization of a Data and Physics-Informed Discrete Loss

In this work by Balcerak et al., the authors present an approach to predicting progression of primary gliomas (GliODIL framework). The authors have devised this framework, which estimates isoclines of tumor cell density surrounding a lesion on imaging using a Fisher-Kolmogorov tumor growth model and optimization of discrete loss, which softly accounts for imaging data and the underlying growth model alongside a PDE constraint. The authors perform their technique on simulated data as well as a test set of 152 glioblastoma patients, a subset of whom have had metabolic imaging with amino acid PET scan.

Overall, I found this work of interest, but am concerned about various aspects of the approach, and note that the claims made by the authors at times are overstated and not necessarily supported by the data presented.

I outline my scientific concerns as below:

- The authors claim on page 2, line 50-51 that FET-PET provides metabolic information that when combined with MRI offers a comprehensive view of tumor behavior. This statement does not appear to be referenced by any source, and I question the degree to which FET-PET can provide additional information for tumor behaviour, as it is a lower resolution study (as the authors note in the results), and there is no validation cited that shows that this actually measures tumor cell density more accurately than MRI.

There are several studies clearly and convincingly linking FET-PET to cellularity and areas of malignancy in gliomas through stereotactic biopsies, i.e., linking histology and the FET-PET signal at voxel level. These include:

- Kunz M, Thon N, Eigenbrod S, et al. Hot spots in dynamic (18)FET-PET delineate malignant tumor parts within suspected WHO grade II gliomas. *Neuro Oncol.* 2011;13(3):307-316.
- Schön S, Cabello J, Liesche-Starnecker F, et al. Imaging glioma biology: spatial comparison of amino acid PET, amide proton transfer, and perfusion-weighted MRI in newly diagnosed gliomas. *Eur J Nucl Med Mol Imaging.* 2020;47(6):1468-1475.

We now cite these works. Thank you for this important comment.

- The motivation for this approach requires careful consideration. The authors seem to imply that a more informed approach to predicting tumor spread and proactively defining margins around the lesion on MRI using a physics-informed model could result in improved outcomes, but this is not at all clear based on the retrospective data presented. For instance, while the authors may accurately predict progression of the tumor with the theoretical model, they do not show any validation data that imply the presence of tumor cells at the time of initial treatment at the predicted margins, and thus, proactive irradiation could result in harm.

In real patients, lacking the ground truth tumor cell distributions known in synthetic cases, defining what to compare against indeed requires careful consideration. The current clinical standard of care in fact consists of uniformly increasing the target volume to a margin of 15mm around the tumor.. This measure is - similar to what we propose - derived from observed tumor recurrences; i.e., it was defined to strike this balance between covering (most) tumor recurrences while not choosing too large an irradiation volume. We therefore feel that comparing against the very same metric - tumor recurrence coverage - is well warranted.

- The authors do not appear to have accounted for the expansile/contractile nature of the brain in their study. In a physics-informed PDE model of glioma growth, particularly for post-surgical cavities, cortical, and subcortical lesions, this omission will likely result in altered results.

Modeling brain deformations and mass effect is indeed a very interesting future research direction. However, the focus of our study was to investigate the potential of GliODIL for RT planning, benchmarking it against a standard-of-care CTV from the current ESTRO-EANO guideline. To allow for an unbiased evaluation, we performed this comparison in the same, preoperative anatomy.

- The authors note on lines 147-148 that the tumor origin may be multifocal and as such, the initial starting point of the tumor was assigned a low importance in the GliODIL solution, but there does not appear to be any analysis of this in the manuscript.

Figure 2 demonstrates how GliODIL is adapted to handle localized multi-focal tumors, including its performance in synthetic studies with cases involving three tumor origins. In contrast, PDE_GliODIL assumes a single focal starting point by forward-running the initial Gaussian function, which results in inferior performance compared to GliODIL, as showcased in Section 2 (Results). For non-localized tumor focal points that are widely spread, our model fails, as highlighted in Figure 7c. This can be addressed in future work. We believe the topic of initial conditions is now thoroughly documented in response to the reviewers' comments. Additionally, Section 5.5 (Initial Guess) discusses the impact of different initial guesses on model performance.

Patient characteristics for 152 GBM patients are not discussed anywhere in the manuscript. These are essential for MRI interpretation. It is also unclear how many had multifocal disease, enhancing disease, median follow-up time, and whether patients had biopsy or surgery. Post-radiation edema is known to differ between MGMT promoter hypermethylated and non-hypermethylated patients, and IDH status also confers unique radiologic properties to tumors - these covariates will also be essential for study interpretation.

Detailed patient characteristics are provided in Section 5.4 (Clinical Dataset), including information on the patient cohort, imaging instruments, and dosages used, as well as tumor and recurrence sizes. This section addresses key aspects relevant to MRI interpretation and provides context for understanding the clinical dataset.

- On various images (e.g. 2E) the model predictions appear to be predicting tumor cell density and edema outside of the boundary of the brain volume – it is unclear to me how this might occur given the stated boundary conditions?

White matter is contoured at 50%, which might give the appearance that tumor cells extend beyond the brain's boundary. However, this does not mean that the cells are outside the brain. In a forward simulation like the one shown in Figure 2E, the boundary conditions ensure that tumor cells remain strictly within the brain volume. In no scenarios did we observe tumor cells outside the brain.

- The analysis on Time Complexity does not appear to be complete. It is an analysis of times for a particular machine to solve the PDE, but not a computational analysis of complexity for each model - this would be useful information, particularly as a function of number of voxels or resolution.

Thank you for the suggestion. We address execution times and spatial resolutions, including computational considerations for each model, in Section 5.6 (Execution Times and Spatial Resolutions). Please refer to this section for a breakdown of the computation times. While we cannot provide per-voxel calculations due to the nonlinear nature of the PDEs, which lack stability guarantees when changing resolutions, we include detailed performance metrics for the current setup.

- It is unclear whether any image normalization or denoising was done on MRI/PET scans before extracting thresholds. How did the authors account for different field strengths and MRI machines?

For MRI processing and segmentation, we relied on well-established pipelines, central to which is the BraTS Toolkit [n1]. This toolkit incorporates multi-center, multi-vendor trained BraTS segmentation algorithms developed using over 1,000 labeled MRI scans, ensuring broad generalizability and robustness. For PET images, attenuation correction was performed according to the vendor's guidelines to normalize signal intensity.

[n1] Kofler et al., BraTS Toolkit: Translating BraTS Brain Tumor Segmentation Algorithms Into Clinical and Scientific Practice. *Frontiers in Neuroscience*, 14, 2020.

Minor issues:

- Line 134: TMCMC has not been defined
- Line 143: 'Spatial' is incorrectly spelled
- Line 201: Apostrophes around "Equal" should be fixed
- Line 397: textbf should be removed from the text
- Equation 10: APET has not been defined

Thank you. It has been corrected. In particular, APET has been renamed to THETA_BKG and mentioned in lines 377-378 / Figure 1c.

We sincerely thank the reviewers for their time, effort, and valuable insights. We have carefully addressed all comments and questions.

Answers in brown.

REVIEWER COMMENTS

Reviewer #1:

The authors have addressed my comments.

Reviewer #2:

The authors have made revisions in response to the reviewers' comments, including providing additional clarifications on the workflow, expanding details on parameter settings, reorganizing the Results section for better readability, which reflect their efforts to improve the manuscript based on the feedback. Some additional suggestions are listed below,

1. While the authors have added an overview of the workflow at the beginning of the Results section, I suggest avoiding the direct repetition of content from Section 4 (Methods). Instead, consider providing a concise summary of the methodology that directly references and aligns with Figure 1, which starts from the input to the output of the model. Additionally, I recommend reorganizing Figure 1 to enhance clarity and coherence. The current layout appears somewhat disorganized, making it challenging to follow the connections between the workflow components. Including clear annotations or labels that explain the equations and their relevance to specific steps within the workflow would further improve the figure's usability.

Thank you for your valuable suggestion. We recognize that the previous layout was challenging to follow due to complex multidirectional arrows and connections. We have now streamlined the workflow overview with a clearer top-down explanation. Additionally, we have thoroughly rewritten the beginning of the Results section (139-171 lines) to align with your recommendation, focusing on explaining the pipeline rather than repeating the Methods section. We believe this revision enhances clarity and improves the flow of information.

2. The optimal weight for λ_{PDE} (set to 1.0) was determined experimentally for the synthetic dataset. Would patient-specific optimization of this parameter improve the model's adaptability and performance?

This is an important question and a potential future direction. The balance between data fidelity and physics priors was determined using noisy synthetic data with broken assumptions rather than patient data, ensuring the validation dataset remains unbiased. If more data becomes available, recurrence cases could be used to further refine the calibration of λ values. This possibility warrants further investigation, and we have added a corresponding clarification in the manuscript (lines 204–207).

3. In the comparison with nnUNET, could you provide more details about the experimental setup? Specifically, what were the raw input features used for the baseline nnUNET experiments, and how were the inputs modified or extended for the nnUNET-FETPET experiments? Additionally, were any specific preprocessing steps applied to ensure consistency between the GliODIL framework and nnUNET results?

Thank you for this question. We have added more details to the description in Section 5.2 Baselines (470 - 480 lines) so that the approach can be more easily reproduced and studied/expanded. The standard nnU-Net experiments used brain tissue distribution maps and pre-operative tumor segmentations as input, while the nnU-Net-FET-PET variant additionally included FET-PET metabolic maps. The network produced binary recurrence segmentations, which we transformed into continuous probability maps using a Euclidean distance-based approach. This conversion

assigns a value of 1 to predicted recurrence regions, with values decreasing smoothly toward 0 as the distance from the predicted areas increases, mimicking tumor cell distribution.

To ensure consistency with GliODIL, we applied the same preprocessing to both methods, including intensity normalization, resampling to a common resolution, and brain masking. The network was trained with an 80/20 train-test split.

Reviewer #3:

Thank you for addressing my edits. Strong revision.

no further comments

Review: Individualizing Glioma Radiotherapy Planning by Optimization 1 of a Data and Physics-Informed Discrete Loss

In this work by Balcerak et al., the authors present an approach to predicting progression of primary gliomas (GliODIL framework). The authors have devised this framework, which estimates isoclines of tumor cell density surrounding a lesion on imaging using a Fisher-Kolmogorov tumor growth model and optimization of discrete loss, which softly accounts for imaging data and the underlying growth model alongside a PDE constraint. The authors perform their technique on simulated data as well as a test set of 152 glioblastoma patients, a subset of whom have had metabolic imaging with amino acid PET scan.

Overall, I found this work of interest, but am concerned about various aspects of the approach, and note that the claims made by the authors at times are overstated and not necessarily supported by the data presented.

I outline my scientific concerns as below:

- The authors claim on page 2, line 50-51 that FET-PET provides metabolic information that when combined with MRI offers a comprehensive view of tumor behavior. This statement does not appear to be referenced by any source, and I question the degree to which FET-PET can provide additional information for tumor behaviour, as it is a lower resolution study (as the authors note in the results), and there is no validation cited that shows that this actually measures tumor cell density more accurately than MRI.
- The motivation for this approach requires careful consideration. The authors seem to imply that a more informed approach to predicting tumor spread and proactively defining margins around the lesion on MRI using a physics-informed model could result in improved outcomes, but this is not at all clear based on the retrospective data presented. For instance, while the authors may accurately predict progression of the tumor with the theoretical model, they do not show any validation data that imply the presence of tumor cells at the time of initial treatment at the predicted margins, and thus, proactive irradiation could result in harm.
- The authors do not appear to have accounted for the expansile/contractile nature of the brain in their study. In a physics-informed PDE model of glioma growth, particularly for post-surgical cavities, cortical, and subcortical lesions, this omission will likely result in altered results.
- The authors note on lines 147-148 that the tumor origin may be multifocal and as such, the initial starting point of the tumor was assigned a low importance in the GliODIL solution, but there does not appear to be any analysis of this in the manuscript.
- Patient characteristics for 152 GBM patients are not discussed anywhere in the manuscript. These are essential for MRI interpretation. It is also unclear how many had multifocal disease, enhancing disease, median follow up time, and whether patients had biopsy or surgery. Post-radiation edema is known to differ between MGMT promoter hypermethylated and non-hypermethylated patients, and IDH status also confers unique radiologic properties to tumors - these covariates will also be essential for study interpretation.
- On various images (e.g. 2E) the model predictions appear to be predicting tumor cell density and edema outside of the boundary of the brain volume – it is unclear to me how this might occur given the stated boundary conditions?
- The analysis on Time Complexity does not appear to be complete. It is an analysis of times for a particular machine to solve the PDE, but not a computational analysis of complexity for each model - this would be useful information, particularly as a function of number of voxels or resolution.
- It is unclear whether any image normalization or denoising was done on MRI/PET scans before extracting thresholds. How did the authors account for different field strengths and MRI machines?

Minor issues:

- Line 134: TMCMC has not been defined

- Line 143: Spatial is incorrectly spelled
- Line 201: Apostrophes around “Equal” should be fixed
- Line 397: `textbf` should be removed from the text
- Equation 10: A_{PET} has not been defined